# Do Hydrothermal Shrimp Smell Vents?

**DOI:** 10.3390/insects12111043

**Published:** 2021-11-20

**Authors:** Juliette Ravaux, Julia Machon, Bruce Shillito, Dominique Barthélémy, Louis Amand, Mélanie Cabral, Elise Delcour, Magali Zbinden

**Affiliations:** 1Laboratoire de Biologie des Organismes et Ecosystèmes Aquatiques (BOREA), MNHN, CNRS-2030, IRD-207, Sorbonne Université, UCN, UA, 7 Quai Saint-Bernard, Bâtiment A, 4e étage, 75005 Paris, France; julia.machon@live.fr (J.M.); bruce.shillito@sorbonne-universite.fr (B.S.); louis.amand@sorbonne-universite.fr (L.A.); melanie.cabral@outlook.fr (M.C.); elise.delcour@yahoo.com (E.D.); 2Océanopolis, Port de Plaisance du Moulin Blanc BP 91039, CEDEX 1, 29210 Brest, France; dominique.barthelemy@oceanopolis.com

**Keywords:** hydrothermal shrimp, olfaction, chemosensory perception, thermal detection, grooming, behavior, antennules

## Abstract

**Simple Summary:**

Chemical senses play a crucial role in mediating fundamental behaviors in most animals, including habitat selection and navigation. In the darkness of the deep sea, do shrimp endemic to hydrothermal vents use these senses to locate active emissions? Here, we examine the olfactory behaviors of two species of vent shrimp and one coastal species for comparison, to determine whether hydrothermal species have functional olfactory capacities and respond to environmental cues. Among these cues, food odors and vent fluid markers (chemicals, as well as temperature) were tested. Such in vivo experiments on deep-sea fauna are challenging to conduct because the animals sampled at depth may suffer from decompression. We, therefore, used dedicated pressurized equipment, and designed experiments at both deep-sea and atmospheric pressure. Vent shrimp groom their olfactory organs similarly to other crustaceans, yet they bear a dense bacterial cover, raising questions about the role of bacteria in shrimp olfaction. Whilst hydrothermal shrimp have been shown to possess a functional sense of smell, none of our test setups allowed us to demonstrate significant attraction to odors. Both hydrothermal species, however, showed attraction to warm temperature emissions, supporting the hypothesis that temperature is a major cue for orientation in the hydrothermal environment.

**Abstract:**

Deep-sea species endemic to hydrothermal vents face the critical challenge of detecting active sites in a vast environment devoid of sunlight. This certainly requires specific sensory abilities, among which olfaction could be a relevant sensory modality, since chemical compounds in hydrothermal fluids or food odors could potentially serve as orientation cues. The temperature of the vent fluid might also be used for locating vent sites. The objective of this study is to observe the following key behaviors of olfaction in hydrothermal shrimp, which could provide an insight into their olfactory capacities: (1) grooming behavior; (2) attraction to environmental cues (food odors and fluid markers). We designed experiments at both deep-sea and atmospheric pressure to assess the behavior of the vent shrimp *Rimicaris exoculata* and *Mirocaris fortunata*, as well as of the coastal species *Palaemon elegans* and *Palaemon serratus* for comparison. Here, we show that hydrothermal shrimp groom their sensory appendages similarly to other crustaceans, but this does not clean the dense bacterial biofilm that covers the olfactory structures. These shrimp have previously been shown to possess functional sensory structures, and to detect the environmental olfactory signals tested, but we do not observe significant attraction behavior here. Only temperature, as a signature of vent fluids, clearly attracts vent shrimp and thus is confirmed to be a relevant signal for orientation in their environment.

## 1. Introduction

For most crustaceans, chemodetection is the dominant sensory modality and plays a crucial role in fundamental behaviors, such as social interactions, foraging and food analysis, and habitat assessment and navigation [1,2]. Chemical signals are detected through the following two distinct pathways: ‘olfaction’ and ‘distributed chemodetection’ [2]. Olfaction is mediated by specific sensilla, called ‘aesthetascs’, which are restricted to the lateral flagella of the antennules, and only contain olfactory receptor neurons that project to the olfactory lobes of the brain [3]. Distributed chemodetection is mediated by sensilla containing both chemoreceptor and mechanoreceptor neurons (bimodal sensilla), and by those that are distributed all over the body surface, but more densely on the antennal appendages, where they are thought to be involved in both taste and smell [1,3,4,5]. The sense of smell results from an intermittent interaction between soluble odorant molecules from the environment and the sensilla of the antennal appendages [6]. Chemodetection in crustaceans has been mainly studied in large decapods, such as lobsters [7,8,9,10,11]; see review in [5], but remains poorly documented in shrimp, and especially in deep-sea species.

Deep-sea shrimp endemic to hydrothermal vents along mid-ocean ridges face the critical challenge of detecting active emissions that are scattered sparsely in an environment devoid of sunlight. During the early stages of life, the larvae are released and dispersed in the water column, and must locate a vent site to settle and, thereby, survive [12,13]. As adults, shrimp must assess their environment for foraging, interacting with congeners and selecting the appropriate microhabitat, in an environment characterized by steep physicochemical gradients [14,15,16]. This certainly requires specific sensory abilities, which are still largely unknown.

Among the sensory modalities, vision has been the most extensively documented in the species *Rimicaris exoculata*, which has highly modified eyes that might detect dim light emitted by hot hydrothermal fluid [17]. Other sensory capacities such as thermosensing and mechanoreception, which may allow hot fluid emissions and their acoustic vibrations to be detected, remain to be addressed. Since active vents are characterized by a substantial release of various chemical compounds [18], chemoreception could be a relevant sensory modality for hydrothermal shrimp. The chemical composition of fluids varies from one site to another, but sulfides, manganese, and iron are among the compounds commonly encountered [18,19]. These chemicals might, therefore, be used as orientation cues, either at short distances of a few meters for unstable compounds, such as sulfide, or at long distances of a few kilometers for stable compounds, such as manganese and iron [13,20]. Preliminary studies of olfaction in hydrothermal vent shrimp were conducted on *Rimicaris exoculata* more than 20 years ago, and provided the first description of the antennal appendages, as well as the first recordings of electrophysiological responses to sulfide in antennal filaments [21,22]. We recently addressed olfaction in alvinocaridid vent shrimp, and in palaemonid coastal shrimp for comparison. Our results regarding the peripheral system (anatomy of the olfactory appendages and detection of stimuli by the sensilla) and the central system (anatomy of the brain) are summarized in Figure 1 [23,24,25,26,27].

The anatomy of the peripheral olfactory system does not have any particular characteristics, compared to coastal shrimp, which could account for the enhanced olfactory perception in hydrothermal vent species (Figure 1, panel 1). The only noticeable differences between hydrothermal and coastal shrimp are the following: (i) the occurrence of pore-like structures in the cuticle of hydrothermal shrimp only, probably facilitating the passage of odorant molecules to the ORNs [25]; (ii) the presence of an unexpected bacterial cover on the antennal appendages of vent shrimp, which can be dense and cover the entire surface, including the olfactory sensilla [23]. The microbial fouling of the chemoreceptor organs is generally controlled by a grooming behavior [28]. In order to remove the material and odorants that have accumulated on or between the aesthetascs, the crustaceans frequently wipe their antennules with setal combs located on their third pair of maxillipeds [28,29]. Both the structural integrity and functional role of the antennules would be impaired by extensive microbial fouling [29]. In the absence of cleaning behavior, increased fouling causes severe damage to the antennules, which may eventually lead to loss of the appendage within 2 weeks [30].

The detection abilities of the antennal appendages were investigated through electroantennography on the vent shrimp *Mirocaris fortunata* (Figure 1, panel 2). While long-distance stimuli, such as manganese and iron, did not cause any response, sulfide triggered electrical signals in the aesthetascs of *M. fortunata*. This would argue for the presence of functional olfactory abilities in *M. fortunata,* and possibly in other vent shrimp species, and support the assumption that vent shrimp are able to detect this compound. However, we also demonstrated that this ability is not specific to hydrothermal shrimp species, since the coastal species *Palaemon elegans* was also proven to detect sulfide. Sulfide is, indeed, known for its toxicity and has been shown to trigger escape responses in the shallow-water caridean species *Crangon crangon* (exposed to 20 μM H_2_S [31]) and *Palaemonetes vulgaris* (exposed to 0.08 mM H_2_S, Sofranko and Van Dover, unpublished data).

The structure of the central nervous system may provide information about the sensory capacities of a species; for example, some species of cave-dwelling peracarid crustaceans have well-developed olfactory neuropils, while visual neuropils are absent, therefore reflecting their life in the dark [32]. However, this is not a general trend, as other cavernicolous peracarids possess small olfactory centers instead [32]. Our results on the brain anatomy of vent shrimp (Figure 1, panel 3) showed that they have rather similar or smaller olfactory lobes compared to coastal species, thus showing no particular characteristics related to their life in the abyssal environment [25,26]. However, they have impressively developed higher integrative centers (hemiellipsoid bodies), which has not been documented so far for blind cave-dwelling species [25,26]. This suggests that they would not rely heavily on olfaction, but have strong capacities to process inputs from different sensory pathways in their integrative centers, which may result in good place memory.

The objective of this study is to observe the following key behaviors of olfaction in vent shrimp species, which could give insights into their olfactory capacities:

1-Grooming behavior. A previous observation of microbial fouling on the antennal appendages of vent shrimp raised the question of the existence and efficiency of grooming behavior. Such a behavior was never observed in *R. exoculata,* and only briefly reported for *M. fortunata* [23]. Here, we report observations and in vivo experiments on *R. exoculata* specimens, to investigate their potential grooming behavior, as well as on the coastal species *P. serratus* for comparison.

2-Attraction to environmental cues (food odors and fluid markers). In order to determine whether sulfide is used as an orientation cue in the vent environment, we performed in vivo experiments on *Rimicaris exoculata* and *Mirocaris fortunata* to identify potential attraction or avoidance behavior. Observations during the maintenance of *M. fortunata* also reported active movement and swimming during feeding, and subsequent aggregation near the food source [33]. We, therefore, used food-related odors as a positive control in our setups for behavioral experiments. We also tested a non-chemical stimulus that is part of the signature of the vents, the temperature. Previous observations reported aggregation behavior close to warm-temperature sources for *R. exoculata* [34] and *M. fortunata* [33], suggesting that temperature could also be a cue for vent shrimp navigation. These observations were made opportunistically during maintenance periods, or experiments that were not initially designed to identify an attraction behavior. Here, we designed experiments to assess the behavior of both *R. exoculata* and *M. fortunata* species, as well as of a coastal species for comparison, towards a warm-temperature stimulus.

Such in vivo experiments on vent fauna are challenging to conduct primarily because animals suffer from decompression during sampling at depths and are difficult to maintain in good physiological conditions. Appropriate pressurized devices were previously developed, and provide a unique opportunity to observe peculiar behaviors on live deep-sea vent shrimp at their natural pressure [35,36]. Furthermore, *M. fortunata* survives decompression quite well when sampled at depths no greater than 2000 m, and can be acclimated at atmospheric pressure for several months [33,37,38], and then tested for behavioral responses under conditions similar to those of shallow-water species. Thus, here, we present experiments at both deep-sea and atmospheric pressure.

## 2. Materials and Methods

### 2.1. Animal Collection

*Mirocaris fortunata* and *Rimicaris exoculata* specimens were collected from Mid-Atlantic Ridge vent sites at depths ranging from 800 m to 3600 m (Figure 2). Shrimp were collected and recovered at deep-sea pressure by using the PERISCOP system, composed of an in situ sampling cell directly clamped onto the nozzle of the submersible’s suction device and an isobaric recovery device [35]. Upon their recovery on the ship deck, the shrimp were transferred in a 20-L pressure vessel (IPOCAMP [36]) operated at in situ pressure (from 8 to 30 MPa), in flow-through mode (20 L h^−1^ flow rate), in order to achieve several behavioral experiments. Alternatively, specimens were immediately dissected and their antennules and antennae were fixed for morphological observations.

Specimens of *M. fortunata* were collected for behavioral experiments at atmospheric pressure from the Lucky Strike vent site (1700-m depth) without isobaric recovery, and maintained on board in 5 to 10-L seawater aquaria at 5–9 °C, at atmospheric pressure (Figure 3). This species tolerates the variations in pressure and temperature relatively well during the ascent, from sites that do not exceed 2000-m depth [37,38,39]. At the end of the cruise, specimens were transferred to the Oceanopolis public aquarium (Brest, France) and acclimated for at least two weeks in 80-L open circuit aquaria with oxygenated seawater at 9 °C, in dark conditions (lights were only switched on for a few minutes per day for cleaning and feeding purposes). Each aquarium contained up to ~50 individuals. Shrimp were first fed with hydrothermal mussels, *Bathymodiolus azoricus*, which were progressively replaced by a nutritive powder for crustaceans (LiptoAqua, Madrid, Spain). After acclimation, shrimp were maintained in their rearing tanks for behavior experiments at the Oceanopolis aquarium, or transferred to the AMEX laboratory (Paris, France) in a 120-L aquarium containing up to 40 individuals, with artificial oxygenated seawater (salinity of 35 g L^−1^, Red Sea Salt, Red Sea, Houston, TX, USA) at 9 °C, in dark conditions, and fed twice a week. For both locations, 25–50 W thermostat heaters set to 25 °C were placed in each aquarium to serve as a hot spot for the shrimp.

*Palaemon elegans* and *Palaemon serratus* (Decapoda, Caridea) are commonly found in coastal waters of the eastern Atlantic and, due to their taxonomic proximity to shrimp species from Atlantic hydrothermal vents, have already been used as models for comparison with deep-sea shrimp in terms of their sensory abilities [23,24,25]. Specimens of *Palaemon elegans* were collected from the Bay of Saint-Malo, using a shrimp hand net (Figure 3). They were transported to the laboratory and transferred to aerated aquaria filled with artificial seawater (salinity of 35 g L^−1^) for at least two weeks of acclimation at room temperature (about 20 °C) under a 12h:12h light:dark cycle, and fed three times a week with shrimp food pellets (Novo Prawn, JBL, Neuhofen, Germany). Specimens of *Palaemon serratus* were collected from Portsall (Brittany, France), maintained for one week in their native seawater and fed with limpets and shrimp pellets (Figure 3). They were then transferred to the laboratory and maintained in the same conditions as *M. fortunata*, except for the temperature, which was set to 10 °C. Three specimens were dissected upon their arrival at the laboratory, and three others after the grooming experiment, and their antennules and antennae were fixed for further morphological observations.

### 2.2. Behavior Experiments at In Situ Pressure on Hydrothermal Shrimp in Pressurized Aquaria 

A total of 15 independent experiments were conducted in the pressurized tanks (Figure 3 and Figure 4), as follows:-Four for grooming (3 batches of *n* = 3 adults and one batch of *n* = 8 juveniles of *R. exoculata*; total *n* = 17);-Two for sulfide pulses (one batch of *n* = 20 *M. fortunata* tested twice, and one batch of *n* = 20 *R. exoculata* tested once; total *n* = 40);-Eight for food and sulfide stimuli (3 batches of *n* = 10 *R. exoculata* for sulfide pH11 and 3 batches of *n* = 10 for sulfide pH4, and 2 batches of *n* = 5 or 6 *M. fortunata* for food; total *n* = 71);-One for temperature pulses (one batch of *n* = 20 adults and *n* = 10 juveniles of *R. exoculata*).

The survival rate was 100% for all experiments, except for the temperature pulse experiment where 2 out of 10 juveniles of *R. exoculata* died during the experiment.

#### 2.2.1. Grooming Behavior in *Rimicaris exoculata*

A total of 9 adults and 8 juveniles of *R. exoculata* were observed for their grooming behavior during the TRANSECT 2018 cruise (Figure 4b). The observations were conducted on 3 batches of 3 adults (1 male and 2 females) and 1 batch of 8 juveniles as follows:-Batch 1: The specimens were collected at the Rainbow site (2300-m depth), and 3 individuals were transferred to IPOCAMP at 23 MPa and 10 °C upon their arrival on board;-Batch 2: The specimens were collected at the Broken Spur site (3100-m depth), and transferred to IPOCAMP at 30 MPa for maintenance at 20 °C. After 7 h, the aquarium was opened to select 3 individuals who were placed in a cage and further repressurized at 30 MPa and 20 °C in another IPOCAMP aquarium for the grooming observations;-Batch 3: The specimens were collected at the Broken Spur site, and transferred to BALIST aquarium at 30 MPa for maintenance at 10 °C. After 48 h, the aquarium was opened to select 3 adult individuals that were placed in a cage and further repressurized in IPOCAMP at 30 MPa and 10 °C for the grooming observations. After 96 h, the BALIST aquarium was opened again to select 8 juvenile individuals who were transferred to IPOCAMP for the same grooming observations as the adults.

For adults of *R. exoculata*, behavior was recorded and analyzed for each individual over a duration of 53 min. The time when the heads of individuals were not visible (individuals in swarms, or on the walls of the cage) was deducted for a more accurate estimation of the number of grooming events per minute per individual. For juveniles, the batch of 8 individuals was very active, which made it difficult to track each individual, especially since they often gathered in swarms. A total of 9 observations were therefore made over a period of 17 min on randomly selected individuals. This gives a total observation time of 153 min (9 × 17 min), comparable to the observation time for *P. serratus* juveniles, which is 159 min (3 × 53 min, see Section 2.3.1). 

#### 2.2.2. Responses to Sulfide Pulse Stimuli on a Batch of *Mirocaris fortunata* and *Rimicaris exoculata*

An experiment at in situ pressure was conducted on *M. fortunata* during the BIOBAZ (2013) cruise to test the attraction to a sulfide pulse (Figure 4d). The shrimp were sampled from the Menez Gwen site (800-m depth) and recovered in the IPOCAMP aquarium at 8 MPa, 9 °C for 1 h (*n* = 20 individuals). Three successive pulses of 10 min with increasing concentrations of sulfide were applied by injecting 3 L of Na_2_S solutions at 25, 50 and 100 μmol L^−1^ via the recirculating seawater system entrance, with one-hour interval between each pulse. These sulfide concentrations include and even exceed the sulfide concentrations measured in the habitat of *M. fortunata* (5.11–38.31 µmol L^−1^ at the Lucky Strike vent site [40]). The shrimp behavior was video recorded during the whole experiment. The number of shrimp in a 6 cm^2^ -surface around the pulse entrance was counted every minute (except when the camera field of view was obstructed by shrimp) for 5 min prior to the beginning of each pulse and until the end of the pulse.

An experiment at in situ pressure was conducted on *R. exoculata* during the BICOSE 2014 cruise to test the attraction to a sulfide pulse. The shrimp were sampled from the Snake Pit site (about 3500 -m depth), and were further stored in the BALIST pressure aquarium [41] at 30 MPa, 10 °C for 38 -h maintenance. A total of 20 adults and 10 juveniles were then transferred to the IPOCAMP aquarium at 30 MPa and 4 °C for the experiment. Three successive 10 -min pulses of increasing concentrations of sulfide were applied by injecting 3 L of Na_2_S solution at 10, 75 and 300 μmol L^−1^ via the recirculating seawater system entrance, with one-hour interval between each pulse. These sulfide concentrations include the sulfide concentrations measured in the habitat of *R. exoculata* (0.5–77 µmol L^−1^ at the TAG vent site; Cathalot C., measurements on board during BICOSE 2018). The shrimp behavior was video -recorded during the whole experiment. The number of shrimp in contact with the pulse entrance (plastic cap outlined with a marker black line) was counted every minute (except when the camera field of view was obstructed by shrimp) for 10 min prior to the beginning of each pulse and 10 min after the end of each pulse.

#### 2.2.3. Response to Food and Sulfide Stimuli during Experiments on a Batch of *Mirocaris fortunata* and *Rimicaris exoculata*

Behavior experiments at in situ pressure were conducted during the BICOSE 2018 cruise, with diverse stimuli (Figure 4c). Agarose gels (0.3%) were prepared with either seawater (control gel), mussel extract (food-related odor gel for *Mirocaris fortunata*), or Na_2_S (2 mmol L^−1^) (sulfide gel for *Rimicaris exoculata*). This sulfide concentration was previously shown to trigger a significant response of *M. fortunata* antennal appendages in electroantennography [25], and we consequently selected this concentration for further behavior experiments using agarose gels. When prepared with no pH adjustment, sulfide solutions are extremely basic (e.g., pH 11 for Na_2_S 2 mmol L^−1^) and the main sulfide species under these pH conditions is bisulfide S^2−^, which is poorly released. The release of sulfide under the Na_2_S form (and H_2_S form in the vent habitat) is greatly enhanced when the pH is forced below the dissociation constant (for H_2_S, pKA = 7.05), using acid for example. To test the release of sulfide in conditions closer to the vent shrimp habitat, a gel was prepared with Na_2_S (2 mmol L^−1^) at pH 4 (acid sulfide gel), and another was prepared with only seawater adjusted to pH 4 (pH control gel), using HCl. The gels were casted in stainless steel tubes (dense enough to sink) previously drilled in 50 positions (to allow the diffusion of chemicals outside the gel; see Figure 4c).

*R. exoculata* were sampled from the hydrothermal sites (TAG, 3600-m depth) with the PERISCOP device (pressurized recovery). On board, the PERISCOP was opened, and shrimp were separated in three batches of *n* = 10. One batch was placed at 30 MPa, 10 °C in the IPOCAMP aquarium for the first experiment, and the other two batches were stored at 30 MPa, 10 °C in the BALIST aquarium for the second and third experiments to be conducted later in IPOCAMP. A first set of experiments was conducted on these 3 batches with sulfide gels at pH 11, and a second set of experiments was conducted on another arrival of shrimp with sulfide gels at pH 4. *M. fortunata* were sampled from the hydrothermal site Snake Pit, 3500-m depth, and two experiments were conducted with food gels on a batch of *n* = 5 and another on a batch of *n* = 6. Two hours of recovery followed the re-pressurization step in IPOCAMP. Then, three agarose gels (two controls, one stimulus) were successively introduced into the tank through an isobaric line (meaning no decompression during the process) with an interval of 45 min. The two control gels were always introduced first. The number of shrimp in contact with the gels was counted for 45 min after the introduction of the gel.

#### 2.2.4. Responses to Warm Temperature Pulses at In Situ Pressure on a Batch of *Rimicaris exoculata*

An experiment at in situ pressure was conducted on *R. exoculata* during the BICOSE 2014 cruise to test the attraction to a temperature pulse (Figure 4d). The shrimp were sampled from the Snake Pit site (about 3500-m depth), and a batch of 20 adults and 10 juveniles was transferred to the IPOCAMP aquarium at 30 MPa and 4 °C for the experiment. Six successive 10-min pulses of decreasing temperatures were applied at 25 °C, 10 °C and 5 °C (each pulse was performed twice) via the recirculating seawater system entrance, with one-hour interval between each pulse. During the pulse, a total volume of 3.3 L of warm water was injected into the pressurized aquarium, which is an open circuit (water flow of 20 L h^−1^), and the body of the vessel was maintained at 4 °C by the external cooling system. The shrimp behavior was video recorded during the whole experiment. The number of shrimp in contact with the pulse entrance (plastic cap outlined with a marker black line) was counted every minute (except when the camera field of view was obstructed by shrimp) for 5 min prior to and 15 min after the beginning of each pulse. Juveniles are not taken into account in the results, as 2 individuals died during the experiment (so *n* = 8), and the maximum number of individuals in contact with the hotspot simultaneously was 2, making the interpretation of the data unreliable.

### 2.3. Behavior Experiments at Atmospheric Pressure on Hydrothermal and Coastal Shrimp

#### 2.3.1. Grooming Behavior in *Palaemon serratus*

After 3 weeks to 1 month of maintenance in the laboratory, 3 batches of 3 *P. serratus* juveniles (2 females and 1 male, 2 females and 1 male, 3 females) were transferred to an IPOCAMP aquarium at atmospheric pressure and 10 °C. The experimental setup in IPOCAMP was the same as for *R. exoculata* (Figure 4b). Behavior was recorded and analyzed for each individual over durations of 53 min.

#### 2.3.2. Responses to Food Stimulus during Two-Choice Experiments on Single Individuals of *Palaemon elegans* and *M. fortunata*

For *P. elegans*, two-choice experiments were performed in a plastic tank (32 × 18 × 18 cm) (Figure 5a,b). A shrimp was placed in the tank filled with 8 L of room -temperature seawater, and left to explore for 5 min before the start of the test. Two small gauze bags were then introduced just below the water surface, one containing shell-less mussels (food source) and the other containing only gauze (lure). The animals were then observed for 10 min (first contact, number and duration of contact with either the stimulus or the control). After each trial, the tank was cleaned and filled with fresh seawater. To examine the role of the antennules in food location behavior, the following two ablations were tested: lateral antennule or both medial and lateral antennule. The ablated animals recovered for one week before being tested. The different trials were each performed on 20 shrimp. Preliminary trials were conducted under red light and dim light to reduce visual cues, but behavior was similar to that observed under ambient light (fluorescent tube). For *M. fortunata*, the same experiments (except for the ablations and the tests with light) were conducted at the Oceanopolis aquarium (Brest, France) in a glass tank (30 × 20 × 20 cm) filled with 8 L of seawater at 10 °C, as described above. Both species were starved for at least 48 h prior the start of the experiment.

#### 2.3.3. Responses to Food and Sulfide Stimuli during Multiple-Choice Experiments on Single Individuals of *Mirocaris fortunata* and *Palaemon elegans*

For *P. elegans*, multiple-choice experiments were conducted in a glass tank (30 × 20 × 20 cm) filled with 8 L seawater at room temperature (Figure 5c,d). Agarose gels (0.5%) were prepared with either seawater (control gel), mussel extract (0.1 g mL^−1^) (food-related odor gel), Na_2_S (2 mmol L^−1^) (sulfide gel) or a mixture of mussel extract and sulfide at the same concentrations (food-sulfide gel). Twenty milliliter gels were casted in the bottom of 50 -mL black tubes (Falcon). For each trial, three control gel tubes and one stimulus gel tube were introduced into the bottom of the tank at each corner, with the opening of the tube facing the center of the tank. For each test, we changed the location of the stimulus tube randomly, to rule out a possible effect due to its place in the aquarium rather than its contents. A shrimp was placed in the tank and its behavior was video-recorded for 30 min for further analysis (first entrance, number and duration of entrance in either the stimulus or the control tube). Each trial used a different individual, and the total number of trials was *n* = 23 (food odor), *n* = 17 (food-sulfide) and *n* = 12 (sulfide). After each trial, the tank was cleaned and refilled with fresh seawater. *For M. fortunata*, the same experiments were conducted at the Oceanopolis aquarium in a plastic tank (30 × 20 × 20 cm) filled with 8 L seawater at 10°C. The food-related odor gel was prepared with shrimp food (0.1 g mL^−1^; Liptoaqua food pellets, Liptosa, Madrid, Spain). Since we were not able to observe any specific behavior with this first trial with a food stimulus (see Section 3.4), we did not pursue other stimuli with this setup for *M. fortunata*. The two species were starved for at least 48 h before the experiment. Each trial used a different individual, and the total number of trials was *n* = 10 for *M. fortunata*.

#### 2.3.4. Responses to Food and Sulfide Stimuli during Two-Choice Experiments on Multiple Individuals of *M. fortunata* and *Palaemon elegans*

Two-choice experiments were conducted at the Oceanopolis aquarium in rearing glass tanks (40 × 40 × 40 cm) filled with 80 L of seawater at 10 °C and containing an aquarium thermostat heater set to 25 °C (Figure 5e,f). One tank contained 16 specimens of *M. fortunata*, and another one contained 8 specimens. Diverse 0.5% agarose gels were prepared with either seawater (control gel), shrimp food extract (0.1 g mL^−1^; Liptoaqua food pellets, Liptosa, Madrid, Spain) (food-related odor gel), Na_2_S (2 mmol L^−1^) (sulfide gel) or a mixture of food extract and sulfide at the same concentrations (food-sulfide gel). The gels were casted in 20 mL cubic molds. For each experiment, a control gel and a stimulus gel were introduced on each side of the tank. The position of the stimulus was randomized for each replica. The shrimp behavior was observed for 30 min, and the number of shrimp in contact with the gel was counted every minute. As for a positive control for this setup, a trial was performed with agarose gels prepared with shrimp food extract on two batches of *P. elegans* (*n* = 9 and *n* = 10) in tanks (50 × 25 × 30 cm) filled with 25 L of seawater at 25 °C.

#### 2.3.5. Responses to Temperature Stimulus during Two-Choice Experiment on Multiple Individuals of *Mirocaris fortunata* and *Palaemon elegans*

Choice experiments were conducted in glass tanks (40 × 40 × 40 cm) filled with 80 L seawater at 9°C containing *M. fortunata* (one batch of *n* = 28 or another batch of *n* = 19) and *P. elegans* (two batches of *n* = 20), at the Oceanopolis aquarium (Figure 5g,h). Seawater was continuously renewed in order to maintain a constant temperature of 9 °C in the tank. Thermostat heaters, covered with foam fixed with plastic collars, were used as a hot spot stimulus. For *P. elegans*, a batch of shrimp was placed in a tank and acclimated for 1 h with no heater. Two heaters (one turned on and set at 25 °C, one turned off) were then introduced into the tank on each lateral side, in an upward position. The mean temperature along the foam surface ranged from 9 to 14 °C when the heater was turned on. The number of shrimp on each heater was counted at different time intervals for 180 min. For each consecutive 180 min trial, the heaters position in the tank was inverted. For *M. fortunata*, two batches of shrimp already present in their rearing tanks were tested, as described above. A total of 6 and 4 trials were conducted on *M. fortunata* and *P. elegans* respectively, and repeated on a second batch for each species. Two trials were extended overnight.

### 2.4. Scanning Electron Microscopy (S.E.M.)

For morphological observations, antennae and antennules (both medial and lateral flagella) of 3 individuals were dissected from *R. exoculata* (adults and juveniles) and *P. serratus* specimens before and after the experiments of grooming observations (Figure 2 and Figure 3). The samples were fixed in a 2.5% glutaraldehyde/seawater solution, and further rinsed and post-fixed in osmium tetroxide 1%. They were then dehydrated using an ethanol series, critical-point-dried with an Emitech K850 or a CPD7501 critical point drying apparatus (Quorum Technologies, Laughton, UK), and platinum-coated in a Scancoat six Edwards sputter-unit or gold-coated with a JEOL JFC-1200 fine coater. Observations were carried out with a scanning electron microscope (Cambridge Stereoscan 260 or Hitachi SU3500), operating at 15 kV.

## 3. Results

### 3.1. Grooming Behavior of Olfactory Appendages in Rimicaris exoculata and Palaemon serratus

In order to observe the possible cleaning behavior of the olfactory organs in *Rimicaris exoculata*, we carried out experiments on a small number of shrimp, with a setup allowing close-up filming (Figure 4b). With these experimental conditions, we were able to observe grooming events in *R. exoculata* for the first time. The frequency of these events was then quantified in adults and juveniles of *R. exoculata*, and in juveniles of *P. serratus* for comparison (Figure 6). Since the experimental conditions differ for several parameters (duration of maintenance before the experiment, temperature and pressure) between the three batches of adult *R. exoculata*, each batch is presented separately. The frequency of grooming for the antennules (A1), expressed as the number of events per min (mean ± s.d.), ranged from 0.14 ± 0.13 to 0.58 ± 0.45 for *R. exoculata* adults, and was 0.27 ± 0.22 for *R. exoculata* juveniles and 0.21 ± 0.12 for *P. serratus* juveniles. For the antennae (A2), the grooming frequency varied from 0.03 ± 0.06 to 0.14 ± 0.16 for *R. exoculata*, and was 0.11 ± 0.08 for *R. exoculata* juveniles and 0.15 ± 0.20 for *P. serratus* juveniles.

There is no significant difference in the frequency of grooming between the antennules and the antennae of adult *R. exoculata* for each of the batches observed (*n* = 3/batch, Student’s test for paired samples, batch 1: *p* = 0.10; batch 2: *p* = 0.15; batch 3: *p* = 0.10). However, there is high inter-individual variability within each batch. There is no difference in the frequency of grooming between the antennules and the antennae in *P. serratus* either (*n* = 9, Student’s test for paired samples, *p* = 0.27). In contrast, there is a significant difference in the frequency of grooming of the antennal appendages for juvenile *R. exoculata* (*n* = 9, Student’s paired sample test, *p* = 0.012), with a grooming frequency of 0.27 ± 0.22 for A1 and 0.11 ± 0.08 for A2. Juvenile *R. exoculata* groomed more (2.5 times more) antennules than antennae.

As the experimental conditions vary for several parameters for each batch of adult *R. exoculata*, as well as for batches of juveniles for the two species, comparisons between the batches for grooming frequency are not reliable, and further experiments are, therefore, required to confirm the impact of the different parameters (duration of maintenance, temperature, pressure, developmental stage). Furthermore, the batches consisted of only three individuals, with high inter-individual variability for grooming frequency, which confirms the requirement for further experiments. For all batches that were maintained for various durations in surface or artificial seawater, i.e., in non-natural conditions, we can suspect an impact on grooming behavior due to the modification of the microbiome and bacterial biofilm on the antennal appendages, which also needs to be confirmed in future experiments.

### 3.2. Fouling of Olfactory Appendages in Rimicaris exoculata and Palaemon serratus

In this study, we observed by SEM the amount of fouling on the antennae and antennules of a coastal shrimp and a hydrothermal shrimp, (i) after their collection from their natural environment and (ii) after an aquarium experiment dedicated to the investigation of their grooming behavior. As expected, our observations revealed that there were no differences between the two types of samples (i.e., before and after the aquarium observation) for both species, with grooming behavior being practiced very regularly by the shrimp. We, therefore, decided to pool the two data sets and consider them as a single batch.

In both species, the inter-individual variability is significant, most likely related to the time elapsed since the last molt and the acquisition of a new bacteria-free cuticle. Antennules and antennae show similar fouling patterns. For *Palaemon serratus*, this ranges from a complete absence of bacteria (Figure 7a,b), through sparse coverage (bacteria are present in low density but relatively homogeneously distributed over the segments, Figure 7c,d), to almost complete coverage of the surface of the antenna and antennule segments (Figure 7e,f). The bacterial morphotypes observed on *P. serratus* are diverse; they have a very fine filament, filamentous bacteria, rods and cocci (see Figure 7d,f). A few diatoms were also occasionally observed among the bacteria. For *Rimicaris exoculata*, the same pattern is observed, from a complete absence of bacteria (Figure 8a), through sparse coverage (bacteria are present as a few patches with a high density of rod or filamentous bacteria; Figure 8c), to almost total coverage of the surface of the antennae and antennule segments (Figure 8e for *R. exoculata*), but also of aesthetascs and non-aesthetascs sensilla (Figure 8b,d,f).

An important difference between *R. exoculata* and *P. serratus* is that, despite the comparable abundant bacterial colonization of the antennal segments, the sensilla (aesthetascs and non-aesthetascs) are almost never covered with bacteria in the coastal shrimp.

### 3.3. Experiments at In Situ Pressure on Mirocaris fortunata and Rimicaris exoculata—Attraction to Sulfide or Food Odor Stimuli

A preliminary experiment was carried out during the BIOBAZ cruise to test the attraction to a sulfide emission in *M. fortunata* (Figure 2 and Figure 4d). A batch of 30 individuals, including 20 adults and 10 juveniles, was placed at in situ pressure in the IPOCAMP aquarium and subjected to a series of 10 min-pulses of 25 to 100 µmol L^−1^ Na_2_S solutions (Figure 9a, the results for the juveniles are not presented on the figure). For all the experiments, a maximum of one adult at a time (5% of the adult batch) and one juvenile at a time (10% of the juvenile batch) came in proximity of the sulfide fluid emission. Each experiment was only repeated twice on the same batch of shrimp, which precludes any robust statistical analysis. However, the behavior of the shrimp did not differ during the pulse when compared to the periods before and after the pulse. It can therefore be concluded that the sulfide emissions did not trigger any attraction in *M. fortunata* in our experimental conditions.

A similar experiment was conducted on *R. exoculata* during the BICOSE 2014 cruise (Figure 2 and Figure 4d), with a batch of 20 individuals being subjected to a series of 10 min-pulses of 10 to 300 µmol L^−1^ Na_2_S solutions (Figure 9b). For all the experiments, a maximum of one individual at a time (5% of the batch) came into contact with the sulfide-diffusing device during the pulse, showing no difference with the behavior observed before and after the pulse. In our experimental conditions, the sulfide emissions did not trigger any attraction in *R. exoculata* either. To further investigate a possible attraction to sulfide, we tested another setup for *R. exoculata* in the IPOCAMP aquarium during the BICOSE 2018 cruise at the Mid-Atlantic Ridge (Figure 2 and Figure 4c). Two control gels and one sulfide-loaded gel were consecutively presented to three batches of 10 individuals. An initial experiment was carried out with a 2 mM sulfide gel at pH = 11, and a second experiment with a 2 mM sulfide gel adjusted to pH = 4 for an enhanced release of Na_2_S (Figure 10a,b, see Section 2 for further explanation). The mean number of times of contact with the control gels and the sulfide-loaded gel was not significantly different between the experiments.

A similar setup was used to test the attraction to a food odor source on *M. fortunata* during the BICOSE 2018 cruise on the Mid-Atlantic Ridge (Figure 2 and Figure 4c). Two control gels and one stimulus gel, loaded with *Bathymodiolus* mussel extract, were presented to two batches of five and six shrimp (Figure 10c). A third batch should have been tested, but the shrimp in this batch were in a poor physiological condition and the third experiment was, therefore, canceled. The mean number of times of contact with the control and the food-loaded gel is similar, and, therefore, does not clearly demonstrate an attraction behavior to the food odor source.

### 3.4. Experiments at Atmospheric Pressure on Mirocaris fortunata and Palaemon elegans—Attraction to Sulfide or Food Odor Stimuli

In order to demonstrate the attraction to a food or sulfide odor source, and, eventually, the role of the antennules for this detection, three different experimental setups were tested (Figure 5a–f).

The first design was a two-choice experiment between a food odor source and a lure of identical appearance (Figure 5a,b). Shrimp starved for 48 h were tested one by one for 15 min with the stimulus and the lure positioned beneath the surface. *P. elegans* specimens showed attraction behavior for the odorous food source, as 45% of the shrimp made first contact with the food, compared to only 20% with the lure, and 30% with neither. The detection of the food odor source was worsened by the selective ablation of the lateral antennules (20% of first contact) or both the lateral and medial antennules (25% of first contact) (both ablated groups significantly differed from the control intact group; Fisher exact test, two-sided, *p* = 0.0007 for the lateral antennule ablated group and *p* = 7 × 10^−8^ for the whole antennule ablated group). The decrease in attraction to a food odor source upon ablation of the antennules of *P. elegans* confirms the chemosensory function of these organs to locate food sources, in agreement with the results of Steullet et al. on spiny lobster [1]. However, the other measured indicators, such as average number and duration of contacts, were similar for both the food source and lure in the non-ablated *P. elegans* group. Thus, this experimental design is not optimal for demonstrating an attraction response to a food odor source in *P. elegans*. The experiment was also inconclusive for *M. fortunata*, for which the majority of intact specimens went to the lure first (65% first contact), so we did not perform consecutive ablation experiments for the vent species.

For the second setup, we used a multiple-choice experiment with three tubes containing control gels and one tube containing a stimulus gel (food or sulfide) (Figure 5c,d). The gels were casted into the bottom of black tubes to exclude visual bias. The tubes were placed on the bottom corners of the tank prior to introduction of the shrimp, and the entrances in each tube were monitored for 30 min. *M. fortunata* specimens explored the tank and tubes, but did not show attraction to the food odor source. Only 20% of the first entrances were in the stimulus tube, the average number of entrances in the stimulus tube was 37%, and the average time spent in the stimulus tube was not significantly different from that spent in the control tubes (Figure 11a). *P. elegans* specimens showed significant attraction behavior towards the food odor source; 61% of the shrimp went to the stimulus tube first, the average number of entrances in the stimulus tube was 67%, and the time spent in this tube, compared to the control tubes, was significantly longer (Figure 11b). We also tested behavioral responses to sulfide on both *M. fortunata* and *P. elegans*. Sulfide did not trigger any noticeable attraction behavior in *M. fortunata* with this setup (data not shown). For *P. elegans*, while sulfide alone did not trigger attraction compared to the controls (Figure 11d), a mixture of food and sulfide attracted shrimp significantly (Figure 11c), suggesting that sulfide is neither an attractant nor a repellent for this species. This result was unexpected, since sulfide is well known to be toxic.

These experiments, on a single individual of *M. fortunata*, were inconclusive for both sulfide and food stimuli. However, shrimp are easily observed detecting and locating food in their rearing tanks at both atmospheric and in situ pressure (i.e., in the AbyssBox [37]). Therefore, in order to get closer to the maintenance conditions and to avoid any stress related to the handling of the individuals, we tested a third experimental setup. The experiments were performed directly in the rearing tanks containing the individuals of *M. fortunata*, at the Oceanopolis aquarium (Brest, France). The shrimp were starved for one week before the experiment, the aquarium thermostat heater was left on, and the water inlet and outlet were closed prior to the experiment. Two gels, one control gel and one stimulus gel (food, sulfide, or food-sulfide mixture), were inserted on each side of the aquaria (Figure 5e,f). The number of shrimp in contact with each gel was monitored over 30 min. The experiment was performed on two batches of shrimp (*n* = 16 and *n* = 8), and each batch was tested twice for each stimulus. This setup did not make it possible to demonstrate a clear attraction of the shrimp to the gel containing the olfactory stimulus (food, sulfide, or food-sulfide mixture). The individuals gathered neither on nor in proximity of one gel. In all the 12 trials, a maximum of 3 shrimp gathered on the stimulus gel, for a period of less than 30 s. Additionally, for three of the trials, no shrimp made contact with either gel. This same setup was also tested on two batches of *P. elegans* (*n* = 9 and *n* = 10), with a food stimulus, and was also inconclusive. The maximum number of shrimp that gathered on the gel was indeed similar for the stimulus and control gels, as follows: 3 individuals for both gels for the first batch (*n* = 9), 5 individuals on the stimulus gel, and 3 individuals on the control gel for the second batch (*n* = 10).

For the three experimental setups presented, the two setups on isolated individuals (hanging bags and choice between four tubes) allowed us to demonstrate an attraction to food in *P. elegans*, and an absence of attraction/repulsion for sulfide. On the other hand, none of the three setups made it possible to demonstrate an attraction behavior in *M. fortunata*.

### 3.5. Attraction to Temperature on Rimicaris exoculata, Mirocaris fortunata and Palaemon elegans

We observed the behavior of the *Rimicaris exoculata* placed in the pressurized aquariums at abyssal water temperature (about 4 °C for 1 h 30 min). In this cold water, the animals moved slowly on all surfaces of the aquarium (bottom, vertical wall and lid). We then subjected the shrimp to a thermal stimulus by diffusing water at 25 °C, 15 °C, or 5 °C into the aquarium in short 10 -min pulses (Figure 12a). Upon initiation of the first pulse at 25 °C, the shrimp rapidly aggregated near the water emission; 50% of the shrimp formed a swarm on the diffuser within 2 min, and up to 65% approximately 7 min later (*n* = 20; Figure 12b). A second pulse at 25 °C, performed 1 h after the first, triggered the same aggregation behavior, with up to 85% of the individuals in the swarm. At the end of the two 25 °C-pulses, the swarms dispersed within 3 to 6 min. The two pulses at 10 °C also caused swarms to form, with up to 45% of the individuals, and the swarms took slightly longer to disperse, i.e., more than 10 min. The pulses at 5 °C did not trigger attraction behavior towards the water emission, with a maximum of 25% of the individuals aggregating around the diffuser, which is not significantly higher than the 0–25% range outside of the pulses.

We conducted choice experiments between two aquarium heaters (one on, set at 25 °C, the other off) on several batches of *M. fortunata* and *P. elegans* in their rearing tanks, at 9 °C in the Oceanopolis aquarium (Brest, France) (Figure 5g,h). The number of shrimp on each heater was counted over a period of 180 min, and then after one night. For *P. elegans*, a maximum of 20% of the individuals went on both the on and off heaters throughout the experiment (Figure 13a). In contrast, 30% of the *M. fortunata* shrimp went on the on heater within 30 min, 50% within 180 min, and up to 80% after one night, whereas the number of specimens never exceeded 5% on the off heater (Figure 13b). Furthermore, when examining the distribution of shrimp on the on heater, *M. fortunata* individuals aggregated on the warmest zone of the heater, for which the surface temperature was measured to be approximately 17 °C. The attraction to warm temperatures was, thus, very clear for hydrothermal species, whereas no attraction was observed for the shallow-water species.

## 4. Discussion

### 4.1. Grooming Behavior and Fouling of the Olfactory Appendages

Antennular and antennal grooming behavior is a stereotyped behavior in crustaceans, in which the first and second pairs of antennae are regularly clasped and wiped by the first and third maxillipeds, respectively. This behavior aims to clear away accumulating debris or chemical water-borne molecules on or between the sensilla, in order to preserve sensory perception [28,29]. The presence of a bacterial cover on the antennal appendages of deep-sea hydrothermal shrimp has led to questions about the existence of cleaning behavior, or, at least, about its efficiency [23,24]. Grooming behavior was first mentioned for the hydrothermal shrimp *Mirocaris fortunata* [23], and here we have shown the existence of this behavior in *Rimicaris exoculata*, both in juvenile and adult stages. Extrapolating to other deep-sea hydrothermal shrimp, we can reasonably assume that they all perform antennal appendage grooming. The bacterial fouling on the olfactory organs of vent shrimp is, therefore, not due to the absence of grooming behavior. Adults of *R. exoculata* showed a similar grooming frequency of sensory organs, compared to the caridean shrimp *Macrobrachium rosenbergii* [42]. Bacterial fouling on the sensory organs of vent shrimp would, therefore, not be due to infrequent grooming behavior.

As described previously, the antennae and antennules of hydrothermal shrimp are indeed, unexpectedly (given the exchange function of these structures) densely covered with bacteria [23,24]. Inter-individual variability may be important, most likely related to the time elapsed since the last molt and the acquisition of a new bacteria-free cuticle. As observed for symbiotic bacteria of the cephalothorax, post-molt individuals are practically devoid of bacterial fouling, whereas, as the next molt approaches, the fouling becomes very dense [43]. Here we confirm the existence of significant bacterial fouling on the antennal appendages of *R. exoculata*, both at the juvenile and adult stage, and even on specimens that were observed to actively clean their antennal appendages. For the coastal species *P. serratus*, we observed similarly abundant bacterial colonization of the antennal segments, with high inter-individual variability, most probably related to the molting cycle. Previous observations on the shallow-water shrimp *Palaemon elegans* have reported less dense bacterial development on olfactory organs, with densities 10- to 100-fold lower than for hydrothermal vent species [24]. Although the density of fouling varies between individuals and species, this leads to the suggestion that bacterial fouling of the olfactory appendages occurs in shrimp in general, and perhaps, even more widely, in all crustaceans.

A striking feature of hydrothermal vent shrimp is the fouling of chemosensory sensilla, and, in particular, the aesthetascs dedicated to olfaction, which has never been observed in shallow-water coastal species (this study, [24]). Aesthetascs sensilla are characterized by a thin cuticle [25,44], permeable to environment chemicals, which will cross the cuticle and bind to olfactory sensory neurons. Non-aesthetascs sensilla, on the other hand, have a thick, non-permeable cuticle, so environmental chemicals only enter via a terminal pore, and bind to chemoreceptors [44]. The observation of bacteria covering most of the surface of some aesthetascs, as well as rods occluding the terminal pore of bimodal sensilla, suggests that the functions of these sensilla are severely, if not completely, impaired in hydrothermal species, which is in accordance with the previous assumption of Barbato and Daniel [29]. An alternative hypothesis is that the bacterial communities of the antennal appendages could emit odorant molecules, or modify the chemical signals of the environment by removing or releasing certain substances, thus affecting the olfactory perception of the host [24].

Taken together, our results showed that grooming behavior is not efficient in removing bacteria from sensory structures. A previous study on brachyuran crabs showed that the gill cleaning behavior removed sediment and algae but not bacteria [45]. Instead of cleaning up bacterial fouling, the purpose of this grooming behavior could be to deposit substances on the sensory organs. Schmidt et al. [46] hypothesized that grooming behavior could be used to spread secretions produced by aesthetasc integumental glands, which would have an antifouling (and/or friction-reducing) role, along the aesthetasc length. The pores of such glands have not been observed on aesthetascs of hydrothermal shrimp, and the presence of dense fouling excludes the hypothesis of grooming to spread an antifouling product. However, Trapido-Rosenthal et al. [47] reported that the transport of molecules could occur through the wall of the aesthetascs, due to their very thin cuticle. We can, thus, hypothesize that grooming here may serve to spread bacterial secretions along the shrimp antennas, or, conversely, shrimp secretions on the bacterial film. These product releases may be related to the sensory functions of shrimp, either directly or indirectly. Indeed, the bacterial biofilm has been proposed to play beneficial roles for shrimp sensory cells, without being directly related to olfaction, such as sulfide detoxification or the production of carbon molecules by chemosynthesis [24].

In conclusion, the high densities of bacteria on the antennal appendages are not due to the absence of grooming behavior, and most likely play a role in shrimp olfaction, whether detrimental or fundamental to this sense, which should be investigated in future studies of the sensory abilities of vent shrimp.

### 4.2. Detection of Environmental Cues (Food Odors and Fluid Markers)

Ecological studies often suggest that the distribution of vent fauna is related to food availability or environmental characteristics, such as temperature and chemical conditions [48,49,50,51,52]. This implies that the vent fauna can detect food sources and environmental cues to locate the vent sites and select their microhabitat. Food-related odors might be major attractants for the scavenger species *Mirocaris fortunata* [50], while odors and features related to vent fluids might be more relevant to the species *Rimicaris exoculata*, which relies on symbiosis with chemosynthetic bacteria for its nutrition (see [53] for review). However, the actual stimuli used by animals to orient themselves in their habitat have not been identified to date, and no clear link has been established between environmental cues and behavioral responses such as attraction or repulsion. Preliminary observations have led to the suggestion that food, sulfide, and temperature might be attractants for hydrothermal shrimp [21,33,34]. Here, we tested different setups to observe the behavioral response of the hydrothermal shrimp *M. fortunata* and *R. exoculata* to these stimuli, as well as of the coastal shrimp *P. elegans* for comparison. The results are summarized in Table 1.

Attraction to a food odor source was significantly demonstrated in *P. elegans* using a two-choice and a multiple-choice experiment, but no conclusive results were obtained for *M. fortunata* with the same setups, nor at in situ pressure, although a food odor is expected to trigger attraction behavior for this species as well. This raises several questions regarding the experimental conditions. Among the challenges of behavioral experiments are designing an appropriate setup, choosing the quality and quantity of stimuli, and the prior preparation of the animals (e.g., acclimation, fasting period). The efficient duration of the fasting period in the vent shrimp *M. fortunata*, to eliminate satiety bias, is unknown. During maintenance in aquaria at Océanopolis (Brest) or at the lab (Paris), this species was fed twice a week, and we arbitrarily chose to fast the animals for one week before the experiments at atmospheric pressure. For the in situ pressure experiments, conducted during oceanographic cruises, such fasting periods were not performed, due to the limited availability and multiple uses of the pressurized aquaria. The physiological state of the shrimp is also of major importance to conduct behavior experiments, and especially for animals collected at depth. However, *M. fortunata* specimens acclimated to atmospheric pressure for several weeks were previously proven to survive and exhibit features of fitness, such as feeding and molting [33]. Experiments at in situ pressure on board have also been previously conducted, and the shrimp in these experiments showed good survival rates and good physiological state after re-pressurization [36]. Finally, different parameters influence foraging, such as odor diffusion dynamics, water flow conditions, or the quality and quantity of the stimuli [54]; for example, it has been shown that flow velocities or stimulus concentrations below or above a certain threshold can interfere with, or even prevent, the localization of odor sources [55]. Thus, a configuration designed and adapted for one species is not necessarily appropriate for another species [54], as observed for *P. elegans* versus *M. fortunata*. For vent species, we tested one additional parameter, which was the presence of conspecifics. Thus, when experimental designs on isolated individuals did not reveal an attraction to food in *M. fortunata*, we carried out tests with groups of individuals, but no attraction was observed and therefore no group effect could be demonstrated. The tests for attraction to sulfide on groups of *R. exoculata*, which naturally live in groups, did not show either attraction or group effects such as copying of conspecifics behavior (see below). 

Sulfide is a signature of the hydrothermal fluid and has been previously proposed as a potential orientation cue for vent animals [21,56]. In addition, sulfide is detected by the antennae of *R. exoculata* [21], and by the antennal appendages of *M. fortunata* and *P. elegans* [25]. We, therefore, selected sulfide as a hydrothermal fluid chemical stimulus for the behavior attraction experiments. Renninger and collaborators [21] reported orientation behavior to a piece of sulfidic rock removed from a chimney in *R. exoculata*, and suggested attraction guidance by sulfide. The results of our experiments at atmospheric and in situ pressure do not support this hypothesis, since we did not demonstrate an attraction for sulfide in either *R. exoculata* or *M. fortunata*. The relevance of sulfide as an orientation cue for *M. fortunata* may be questionable, since this species lives at distance from the vent exits, where hydrothermal fluids originate from diffuse seafloor fissure emissions, at low concentrations (e.g., at the Lucky Strike site: 2.4–38 μmol L^−1^ in *M. fortunata* habitat [40,51], 2.5–3 mmol L^−1^ in pure fluid [57]). Furthermore, *M. fortunata*, which does not show a strong association with symbiotic bacteria, is rather an opportunistic feeder, and is, thus, not directly dependent on hydrothermal fluid emissions [50,58]. In contrast, the hydrothermal species *R. exoculata* exhibits a strong dependence on hydrothermal fluid to supplement its symbiotic bacteria with reduced compounds, such as hydrogen sulfide [59]. The lack of an attractive response to the sulfide stimuli was, therefore, more unexpected in this species. No repulsive response was observed in either the two hydrothermal shrimp or the coastal shrimp. However, we cannot conclude with certainty that the chemical components of fluids do not serve as a signal for orientation in hydrothermal species, as this could be due to inappropriate experimental setups, as discussed previously for food odors. The behavioral responses to sulfide were tested on *P. elegans*, with a multiple-choice setup, and, although no attraction to sulfide was observed, the individuals were significantly attracted to a mixture of food odor and sulfide, suggesting that sulfide is not repulsive to this species. Since sulfide is well known for its toxicity, this result was unexpected and was in contradiction with previous results on other coastal shrimp ([31], Sofranko and Van Dover, unpublished data). The ecological relevance of the lack of sulfide repulsion in *P. elegans* is unclear.

Habitat selection in marine species is largely determined by thermal conditions (see, for example, [60]). Deep-sea hydrothermal vents exhibit a very peculiar thermal environment, due to the chaotic mixing of hot hydrothermal fluids with cold abyssal seawater, resulting in thermal regimes characterized by variations of high magnitude over short spatial and temporal scales (see, for example, [61,62]). Temperature is a major marker of active vent sites, and hydrothermal species that live close to the fluid emissions may use this cue to select a suitable microhabitat, as well as to avoid exposure to deleterious temperatures. Previous in vivo experiments reported attraction of *R. exoculata* to a source of warm water (attraction to 11 °C in 3 °C seawater background [34]). For the vent shrimp *Mirocaris f**ortunata*, its relation to temperature has been examined in terms of temperature preference (19.2 ± 1.1 °C [63]), and observations in rearing tanks at in situ and atmospheric pressure showed aggregation behavior on heaters [33]. Nevertheless, attraction to warm temperature, as an adaptive behavior to the vent habitat, has not been robustly characterized in vent shrimp. Our current results from experiments at in situ pressure have clearly shown attraction of *R. exoculata* to warm-temperature emissions. We also showed, with an appropriate setup, including controls, significant attraction behavior of the vent shrimp *M. fortunata* to a warm spot, whereas the same setup did not trigger an attraction response in the coastal shrimp *P. elegans*. Warm temperature elicited attraction behavior in both hydrothermal vent shrimp, but not in the coastal shrimp, supporting the assumption that temperature is a major cue for orientation in the hydrothermal environment. However, the behavioral responses to temperature stimuli were very different for the two hydrothermal species in terms of time scale. Indeed, the *R. exoculata* specimens were instantly attracted to the warm water source and aggregated within minutes, whereas the *M. fortunata* specimens aggregated over longer time scale, within a range of hours (50% of the individuals aggregated on the warm source within 2 to 3 h, and 80% after one night). The *R. exoculata* specimens showed positive thermotaxis behavior, while the behavior of *M. fortunata* more closely resembled thermal preferendum selection. This might be due to the difference in the experimental setup, and, in particular, the baseline temperature in the tank. For *R. exoculata*, the baseline temperature was 4 °C, which is close to the abyssal water temperature, while it was 9 °C for *M. fortunata*, which is closer to the habitat temperature of this species. Further characterization of the thermotaxis kinetics of these species will be necessary to confirm the observed differences, which may be related to their different habitats. *R. exoculata*, indeed, colonizes the walls of hydrothermal vents, close to the fluid outlets, and maintains a close link with the fluids to feed its symbionts. On the other hand, *M. fortunata* lives in a more peripheral zone and does not depend directly on fluid emissions, since it is a secondary consumer. Taken together, our results support the use of temperature for habitat selection in hydrothermal species, and imply that they have mechanisms to sense temperature to effectively exploit their thermal environment, as is generally accepted for many crustaceans [64]. The thermosensitivity of vent shrimp remains to be determined, and, in particular, the potential ability to detect fine temperature variations. Jury and Watson [65] found that lobsters can detect temperature changes exceeding 1 °C, and probably as small as 0.15 °C, by recording changes in heart rate during exposure to thermal variations. In larvae of the crab *Rhithropanopeus harrisii*, vertical migration in the water column is triggered by temperature changes from 0.29 to 0.49 °C [66]. In the area where the hydrothermal fluid mixes with the surrounding seawater, the temperature can vary abruptly from 2 °C to over 30 °C [67]. At the periphery of the rising hydrothermal plume, and in the buoyant hydrothermal plumes spreading over 100 m, the temperature differences with seawater can be less than 0.03 °C [20,68]. Being able to sense such small temperature anomalies could be of major utility for vent shrimp, to detect an active site from a distance.

## 5. Conclusions

Here, we have shown that hydrothermal vent shrimp groom their sensory appendages similarly to other crustaceans. However, this does not eliminate the dense bacterial biofilm that covers the entire antennae, including the olfactory sensilla, thereby raising questions about the role of grooming behavior and, by extension, the role of biofilm in shrimp olfaction. Although shrimp have previously been shown to possess functional sensory structures, and to be able to detect the environmental olfactory cues tested, none of them triggered significant attraction behavior. Only temperature, as a signature of vent fluids, attracted the vent shrimp, and, thus, was confirmed to be a relevant signal for orientation in their environment. Thermosensitivity will need to be investigated further, as well as other tracks, such as mechanoreception, and attraction to congeners. The latter could be of significant importance in *Rimicaris exoculata*, since this species lives in swarms, as a result of social interaction.

## Figures and Tables

**Figure 1 insects-12-01043-f001:**
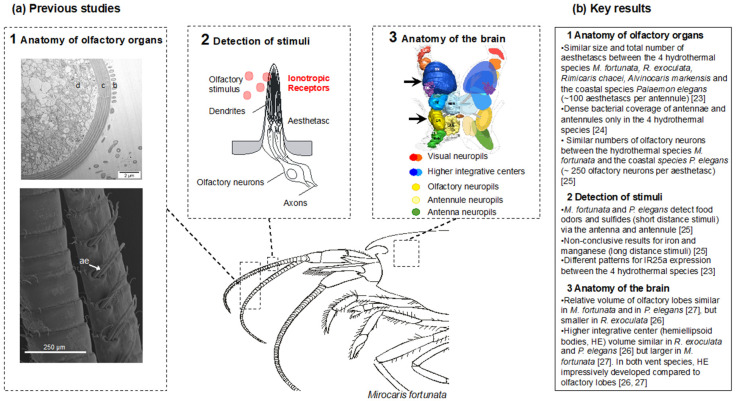
State of the art on hydrothermal shrimp olfaction. (**a**) Previous studies conducted by our team on olfaction in hydrothermal shrimp are summarized in this figure. Panel 1 shows the anatomy and ultrastructure of the olfactory organs of *M. fortunata*. The lower image shows the antennule, which consists of a median flagellum (**left**) and a lateral flagellum with aesthetasc sensilla (**right**). The upper image shows a cross section of an aesthetasc with the cuticle (c), bacteria (b) on the outer surface and dendritic segments (d) of olfactory neurons inside. Panel 2 shows a scheme of an aesthetasc sensilla from a marine crustacean decapod. Detection of various ecologically relevant chemical stimuli (at short and long range) by the antennal appendages was measured by electroantennography. Several ionotropic receptors, including the IR25a co-receptor putatively involved in olfaction, were identified and shown to be mainly expressed in the lateral flagellum of the antennules bearing the aesthetascs. Panel 3 shows a schematic representation of the organization of crustacean brain in dorsal view. Olfactory neuropils (black arrow) receive the sensory input from olfactory neurons that innervate the aesthetasc sensilla. (**b**) Key results from studies on olfactory organs anatomy and stimuli detection [23,24,25], and on brain anatomy [26,27].

**Figure 2 insects-12-01043-f002:**
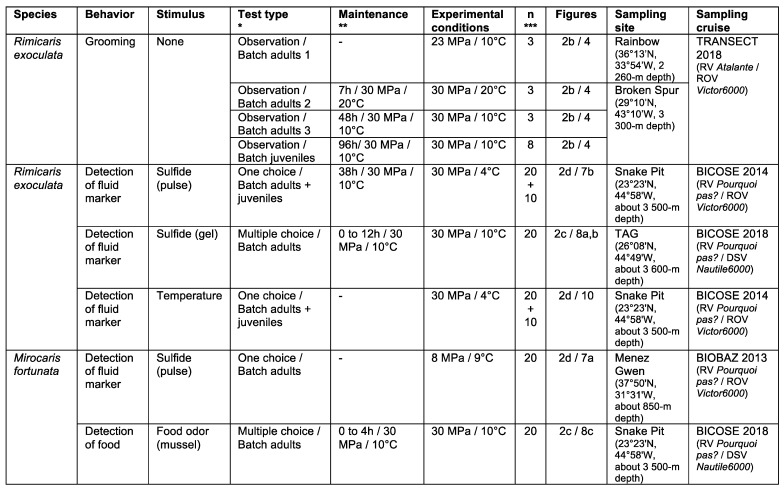
Species collection and experimental condition for behavioral experiments at in situ pressure. * Type of test applied either to a batch of shrimp or to single specimens; ** duration and condition of maintenance in pressurized aquaria before the experiment; *** number of shrimp for each experiment (some experiments were performed several times, see figure legends for more information).

**Figure 3 insects-12-01043-f003:**
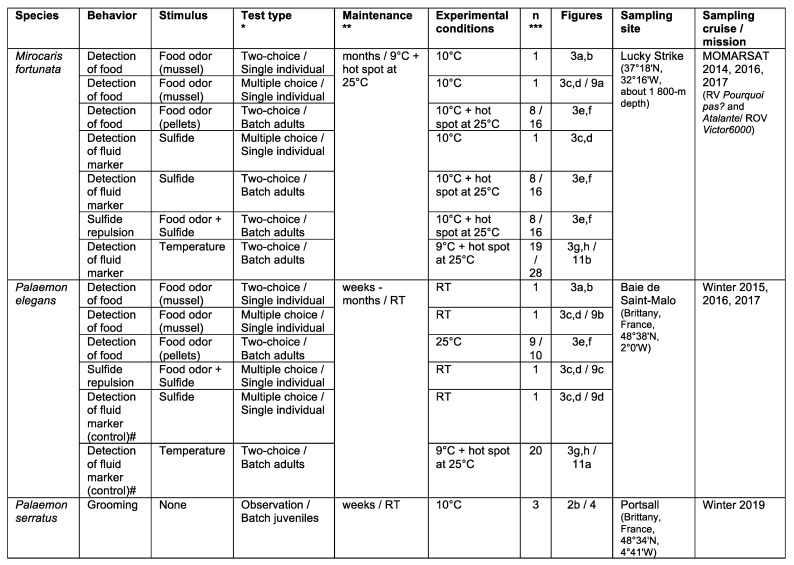
Species collection and experimental condition for behavioral experiments at atmospheric pressure. * Type of test applied either to a batch of shrimp or to single specimens; ** duration and condition of maintenance at atmospheric pressure before the experiment; *** number of shrimp for each experiment (some experiments were performed several times, see figure legends for more information); # experiments with sulfide and temperature were carried out for comparison with hydrothermal species.

**Figure 4 insects-12-01043-f004:**
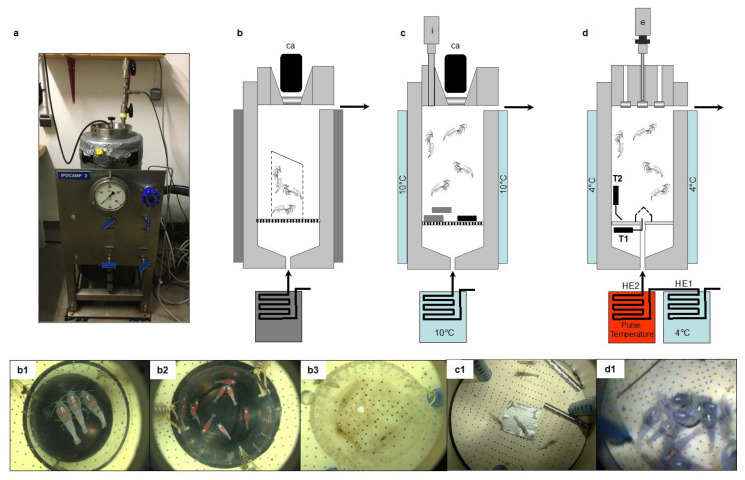
Setups for experiments in the pressure vessel IPOCAMP. (**a**) The pressure vessel IPOCAMP (internal diameter 20 cm, height 60 cm [36]); (**b**) experimental setup for grooming observations. Arrows indicate the inlet and outlet of circulating seawater. This device also used for chemical stimuli experiments, see (**c**), is a recently upgraded version of the lid of the vessel IPOCAMP. The new lid comprises a large viewport and consequently allows direct observation as well as video recording with a high definition camera (ca, AG-HCK10G HD camera head, AG-HMR10 portable recorder, Panasonic). Three shrimp were placed in a PVC cage closed at the top with a transparent polyethylene lid. The behavior of the shrimp was recorded throughout the experiment. Pictures (**b1**–**b3**) are views of the following animals in the IPOCAMP aquarium during grooming observations: (**b1**) *R. exoculata* females and male (marked with a black line); (**b2**) *R. exoculata* juveniles; (**b3**) *Palaemon serratus* juveniles. Body length of the observed shrimp is about 4–5 cm for *R. exoculata* and *P. serratus* juveniles, and about 2 cm for *R. exoculata* juveniles. (**c**) Experimental setup for chemical stimuli experiments. Arrows indicate the inlet and outlet of circulating seawater. The seawater inlet pipe passes through a thermostatically controlled bath. The lid is equipped with an isobaric line (i) that allows the introduction of small elements (e.g., food, stimulus) without disrupting the pressure inside the aquarium. During the experiment, two control gels (grey bars) and one stimulus gel (black bar) were introduced into the tank through the isobaric line at 45 -min intervals. Picture (**c1**) shows *Mirocaris fortunata* in the IPOCAMP aquarium during this experiment. Specimens lie on the plastic bottom of the tank with holes for water entrance, two stainless steel tubes that contain gels are visible. (**d**) Experimental setup for temperature and sulfide pulse experiments. Three sapphire viewports in the pressure vessel lid allow the insertion of an endoscope (e) (Fort, Dourdan, France) and two optical -fiber light -guides for the behavioral observation. The experiments are recorded by a CCD camera (JVC, TK-C1380) and a DVD recorder (DVO-1000MD, Sony). A diffuser system, consisting of a plastic cap with holes, is placed over the seawater inlet hole. Warm -temperature pulses were obtained by immersing a heat exchanger (HE) located on the water inlet line in a temperature -controlled water bath that had been preset to the desired pulse temperature. Two Pt-100 autonomous temperature loggers (S2T6000D, NKE Instruments) are positioned in the upstream water flow (T1) and on the plastic plate at the bottom of the tank several centimeters (>5 cm) from the hot water diffuser (T2). For the sulfide pulse experiments, the HE2 and temperature loggers were removed, and the sulfide solutions were injected into the seawater inlet pipe. Picture (**d1**) shows *Rimicaris exoculata* in IPOCAMP during a temperature pulse experiment at 25 °C. Specimens gathered around the warm water diffuser (marked with a dark circle).

**Figure 5 insects-12-01043-f005:**
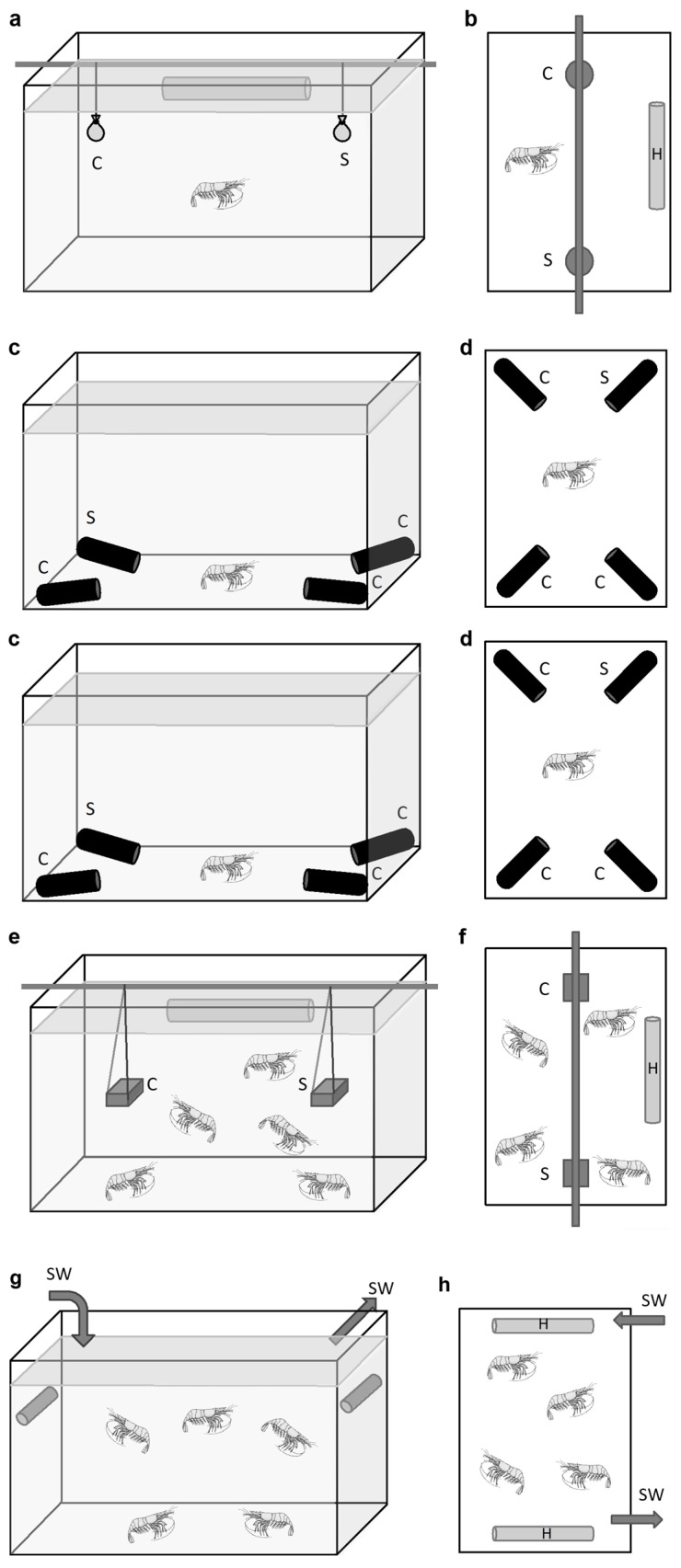
Experimental setups for experiments in aquaria at atmospheric pressure. Perspective (**a**) and birds eye (**b**) views of the experimental setup for two-choice experiments on single *P. elegans* and *M. fortunata*. The shrimp was placed in the aquarium (*P. elegans*, room temperature; *M. fortunata*, 10 °C) and left to explore for 5 min. Two gauze bags (one control (C), one stimulus (S)) were introduced on each side of the tank. The shrimp were observed for 5 min. Perspective (**c**) and birds eye (**d**) views of the experimental setup for multiple-choice experiments on single *P. elegans* and *M. fortunata.* The shrimp was placed in the aquarium (*P. elegans*, room temperature; *M. fortunata*, 10 °C) after the introduction of four tubes containing agarose gels (three controls (C), one stimulus (S)) in the corners of the tank. The behavior of the shrimp was recorded for 30 min using a camera placed above the aquarium. Perspective (**e**) and birds eye (**f**) views of the experimental setup for two-choice experiments on a batch of *M. fortunata*. The experiments were conducted on several individuals in rearing tanks at 9 °C containing a heating thermostat (H) set at 25 °C. Two gels (one control (C), one stimulus (S)) were introduced on each side of the tank. The shrimp were observed for 30 min. Perspective (**g**) and birds eye (**h**) views of the experimental setup for experiments of choice between ON and OFF thermostats on a batch of *P. elegans* and *M. fortunata.* The experiments were conducted in rearing tanks at 9 °C containing several individuals of *P. elegans* or *M. fortunata.* Two temperature thermostats (one ON, the other OFF) were placed on the lateral sides of the tank, close to the surface. The number of shrimp positions on each thermostat was counted for 180 min, and then after one night. Arrows indicate the inlet and outlet of circulating seawater (SW).

**Figure 6 insects-12-01043-f006:**
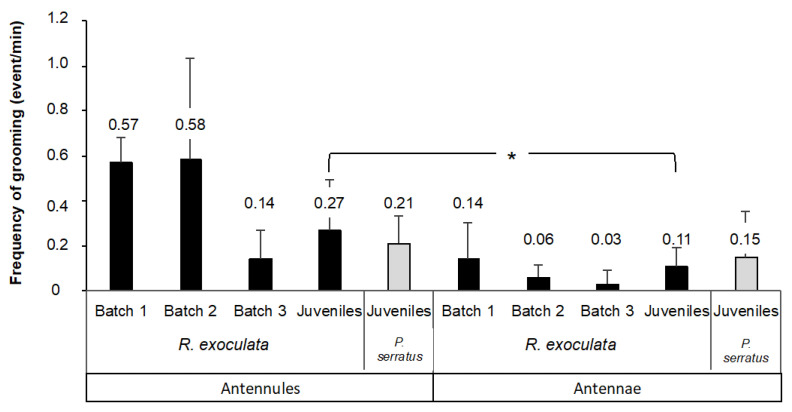
Frequency of grooming behavior of olfactory appendages in *Rimicaris exoculata* and *Palaemon serratus*. Frequencies are expressed as the number of grooming events per min (mean ± S.D.) for each batch of individuals for adult specimens of *R. exoculata* (*n* = 3) and for the 3 batches of individuals for juveniles of *P. serratus* (*n* = 9). The observations were carried out over a period of 53 min for each batch, and observations for the 3 batches of *P. serratus* juveniles were pooled, therefore corresponding to a total observation time of 159 min. For *R. exoculata* juveniles (*n* = 8), 9 observations were carried out over a period of approximately 17 min on randomly chosen individuals, corresponding to a total observation time of 153 min. The experimental conditions are as follows: (1) *R. exoculata* adult batch 1: direct observation, 23 MPa, 10 °C; (2) *R. exoculata* adult batch 2: 7 h maintenance before observation, 30 MPa, 20 °C; (3) *R. exoculata* adult batch 3: 48 h maintenance before observation, 30 MPa, 10 °C; (4) *R. exoculata* juveniles: 96 h maintenance before observation, 30 MPa, 20 °C; (5) *P. serratus* juveniles: 3 weeks to 1 month maintenance before observation, atmospheric pressure, 10 °C. * Significant difference in the frequency of grooming (*n* = 9, Student’s paired sample test, *p* = 0.012).

**Figure 7 insects-12-01043-f007:**
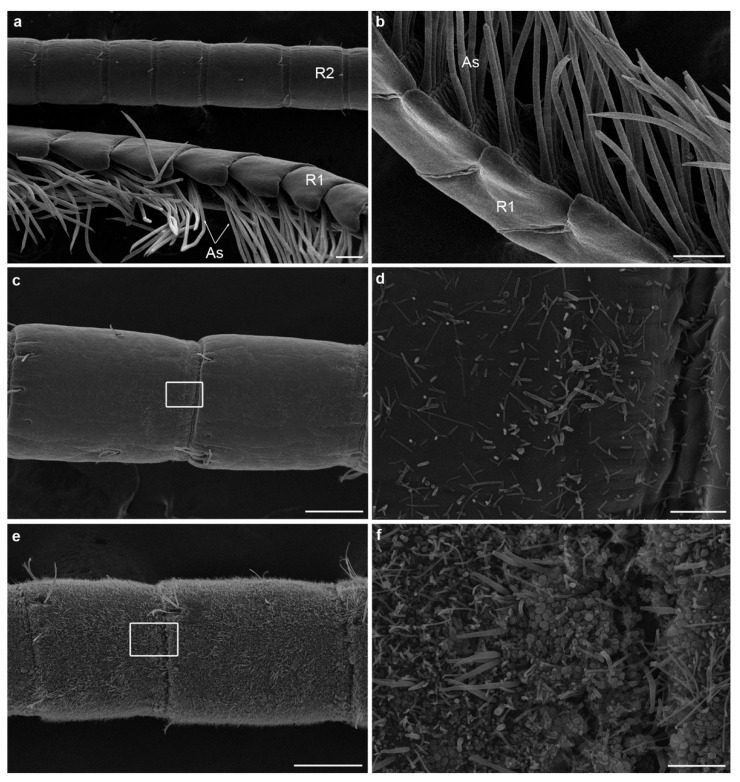
Scanning electron micrographs of *Palaemon serratus* (juveniles) before (Ps1, Ps2) and after (Ps10, Ps11) grooming experiments. (**a**,**c**,**e**) Antennules of Ps10 (**a**), Ps1 (**c**) and Ps2 (**e**) specimens showing gradation from absent (**a**), to moderate (**c**) and intense (**e**) bacterial fouling on the antennal segments. (**a**) Antennules of Ps10 specimen, showing the two ramus and the aesthetascs completely devoid of bacterial fouling; (**b**) close-up on the aesthetascs of Ps11 specimen completely devoid of bacterial fouling; (**c**,**d**) antennae of Ps1 specimen with a light fouling of bacteria. Frame in (**c**) is enlarged in **d** showing the bacterial morphological variety, with thick and thin filamentous bacteria, some rods and cocci; (**e**,**f**) antennules of Ps2 specimen with very dense fouling covering the entire surface of the segments. Frame in (**e**) is enlarged in (**f**), showing bacterial density and morphological diversity. As: aesthetascs, R1 and R2: the two rami of the lateral antennular flagellum. Scale bars: (**a**–**c**,**e**) = 100 µm; (**d**,**f**) = 10 µm.

**Figure 8 insects-12-01043-f008:**
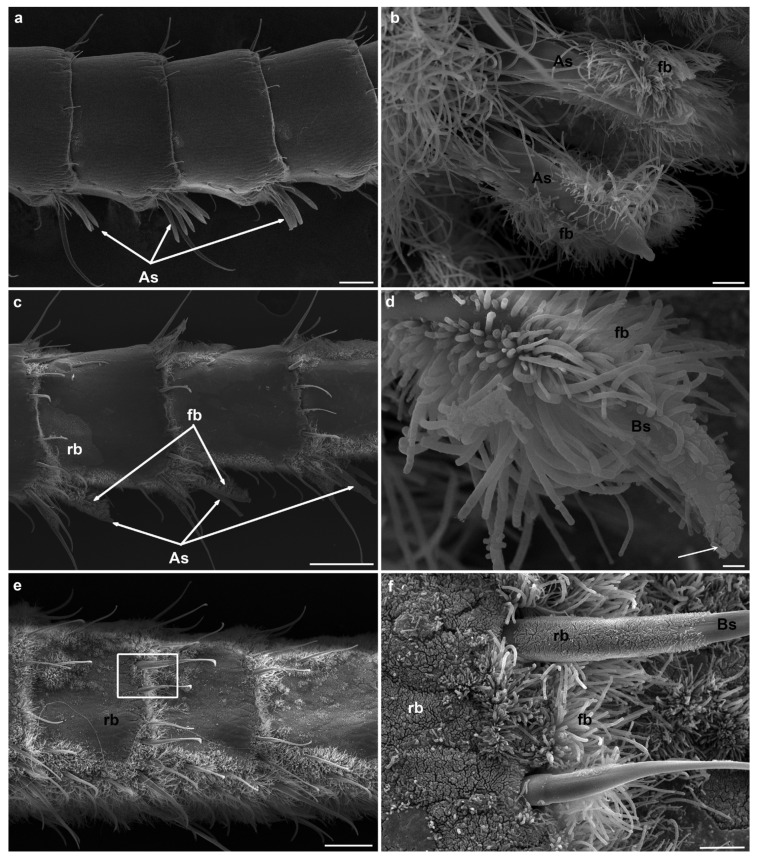
Scanning electron micrographs of *Rimicaris exoculata* before (Rex1, Rex5) and after (Rex38) grooming experiments. (**a**,**c**,**e**) Antennules of Rex1 (**a**), and Rex5 (**c**,**e**) specimen showing gradation from no, light and high bacterial fouling on the antennal segments; (**b**,**d**,**f**) Antennule of Rex38 specimen showing dense bacterial fouling on the aesthetascs (**b**) and beaked setae (**d**,**f**) (terminology used in [23]) including their terminal pore. In (**d**) arrow shows the location of the pore covered by rods). Frame in (**e**) is enlarged in (**f**), showing the bacterial (rods) mat covering the surface of the segments, the filamentous bacteria (fb) in the inter-segmental areas and the base of a short thin setae, as well as and an intermediate beaked setae covered by rod-shaped bacteria. As: aesthetascs, Bs: beaked seta, fb: filamentous bacteria, rb: rod-shaped bacteria. Scale bars: (**a**,**c**,**e**) = 100 µm; (**b**,**f**) = 10 µm; (**d**) = 2 µm.

**Figure 9 insects-12-01043-f009:**
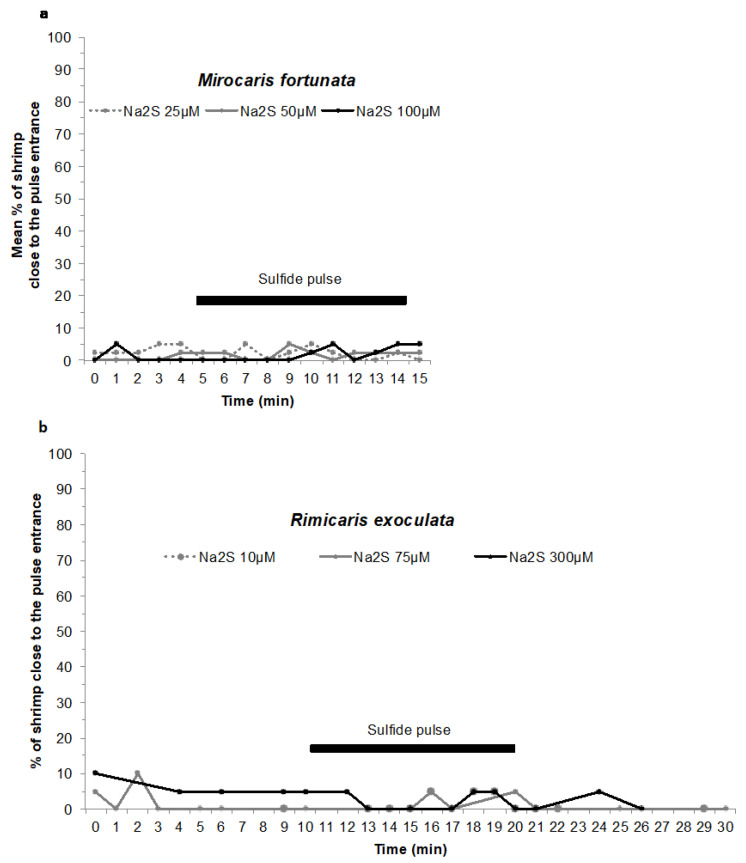
Responses to sulfide pulse stimuli at in situ pressure on a batch of *Mirocaris fortunata* and *Rimicaris exoculata.* (**a**) *Mirocaris fortunata*. Mean % of shrimp over an area of 6-cm^2^ surface around the seawater inlet hole. Three injections were carried out with increasing concentrations of sulfide solutions (25, 50 and 100 µM), and these sulfide pulses are indicated by a black bar along the time scale. One-hour interval separates each sulfide pulse. The experiment was conducted twice on the same batch (*n* = 20) of shrimp (i.e., each point represents the mean of two replicates). The experimental setup is depicted in Figure 4c, with the following modifications: removal of the HE2, the two temperature loggers and the diffuser plastic cap; (**b**) *Rimicaris exoculata.* % of shrimp in contact with the diffuser. Three injections were carried out once on the same batch of shrimp (*n* = 20), with increasing concentrations of sulfide solutions (10, 75 and 300 µM). The number of shrimp close to the pulse entrance was 0 for the experiment with 10 µM Na2S, and thus the result line is merged with the x-axis. A black bar along the time scale indicates these sulfide pulses. One-hour interval separates each sulfide pulse. The experimental setup is depicted in Figure 4c, with the following modifications: removal of the HE2 and the two temperature loggers.

**Figure 10 insects-12-01043-f010:**
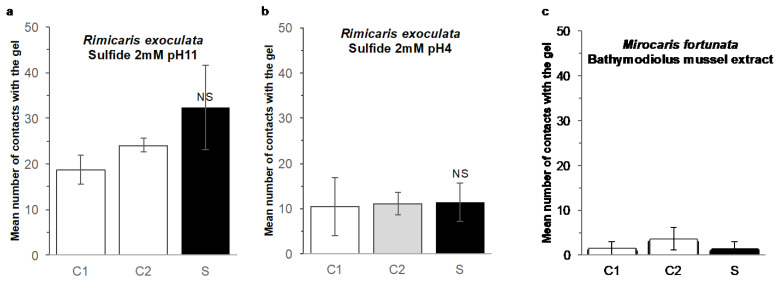
Response to food and sulfide stimuli during experiments at in situ pressure on a batch of *Rimicaris exoculata* and *Mirocaris fortunata.* The shrimp were placed in the IPOCAMP aquarium for a recovery period of 2 h at 30 MPa and 10 °C (see Figure 4c for the setup description). Three gels (2 controls, 1 stimulus) were then introduced consecutively through an isobaric line with an interval of 45 min. The number of times of contact of the shrimp with the newly introduced gel was quantified over a period of 45 min (**a**) *Rimicaris exoculata.* Mean number of times of contact (±S.E.M.) with the stimulus (S, agarose gel loaded with a 2mM sulfide solution at pH11) and the control gels (C1 and C2, agarose gels). Three batches of 10 shrimp were tested once. The mean number of times of contact with the stimulus gel was compared to those of control gels with a two-tailed t-test for correlated samples (same batch of shrimp under different test conditions) (df = 2), and was not significantly different (NS) (S/C1, *p* = 0.068; S/C2, *p* = 0.098). (**b**) *Rimicaris exoculata.* Mean number of contacts (±S.E.M.) with the stimulus (S, agarose gel loaded with a 2mM sulfide solution at pH 4, black bar) and the control gels (C1, agarose gel, white bar; C2, pH 4 agarose gel, light grey bar). Three batches of 10 shrimps were tested once. The mean number of times of contact with the stimulus gel was compared to that of control gels with a two-tailed t-test for correlated samples (df = 2), and was not significantly different (NS) (S/C1, *p* = 0.901; S/C2, *p* = 0.965). (**c**) *Mirocaris fortunata*. Mean number of times of contact (±S.E.M.) with the stimulus (S, agarose gel loaded with mussel extract, black bar) and the control gels (C1 and C2, agarose gels, white bars). Two batches of 6 and 5 shrimp were tested once, and no statistical analysis was performed (*n* = 2).

**Figure 11 insects-12-01043-f011:**
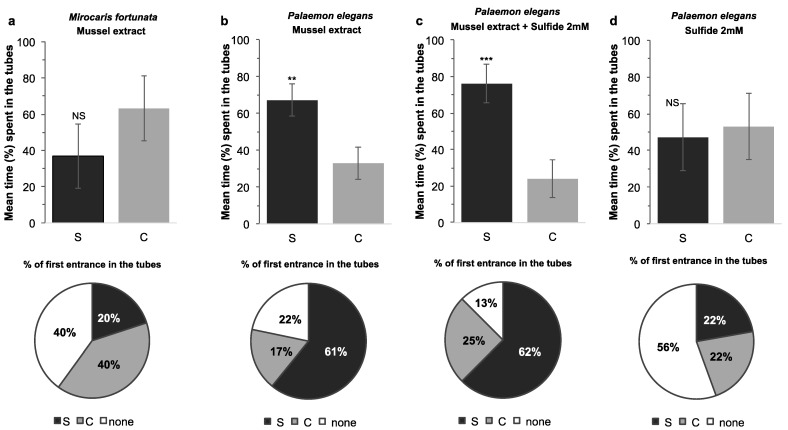
Responses to food and sulfide stimuli during multiple choice experiments on single individuals of *Mirocaris fortunata* and *Palaemon elegans*. (**a**) *Mirocaris fortunata.* Upper graph (bar chart): mean time (% ± S.E.M.) spent in the tube containing the stimulus odor (S, agarose gel loaded with mussel extract, black bar) and in the 3 control tubes (C, agarose gels, grey bar). The means were compared with a two-tailed t-test (not significantly different, *n* = 10). (**b**–**d**) *Palaemon elegans*. Upper graphs (bar charts): mean time (% ± S.E.M.) spent in the tube containing the stimulus odor (S, agarose gel loaded with mussel extract ((**b**), *n* = 23), mussel extract + sulfide 2 mM ((**c**), *n* = 17), sulfide 2 mM ((**d**), *n* = 12), black bar) and in the 3 control tubes (C, agarose gels, grey bar). The means were compared with a two-tailed t-test ((**b**), *p* = 0.008, ** *p* < 0.01, significantly different; (**c**), *p* = 0.001, *** *p* < 0.005, significantly different; (**d**), not significantly different). Lower graphs (pie charts) present the distribution of the shrimp according to their first entrance in the stimulus tube (S, black), a control tube (C, grey) or none (white).

**Figure 12 insects-12-01043-f012:**
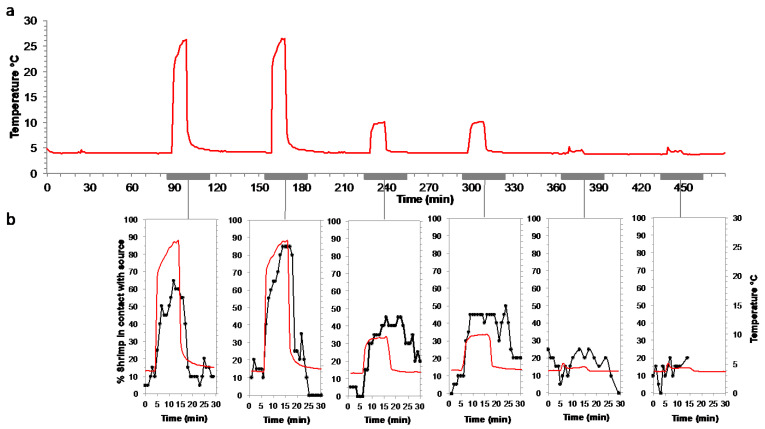
Responses to warm-temperature pulses at in situ pressure on a batch of *Rimicaris exoculata.* (**a**) Profile of warm temperature pulses. The baseline temperature was set at 4 °C, the first two pulses were set at 25 °C, the next two pulses at 10 °C and the last two pulses at 5 °C, with one hour between each pulse; (**b**) percentage of shrimp (*n* = 20) in contact with the diffuser for each pulse. The temperature profile is plotted for each 30 min observation period (symbolized by a grey bar on the temperature profile in (**a**)). For the last temperature pulse, observations after 15 min are missing due to the poor quality of the video (camera field of view obstructed by shrimp and low light).

**Figure 13 insects-12-01043-f013:**
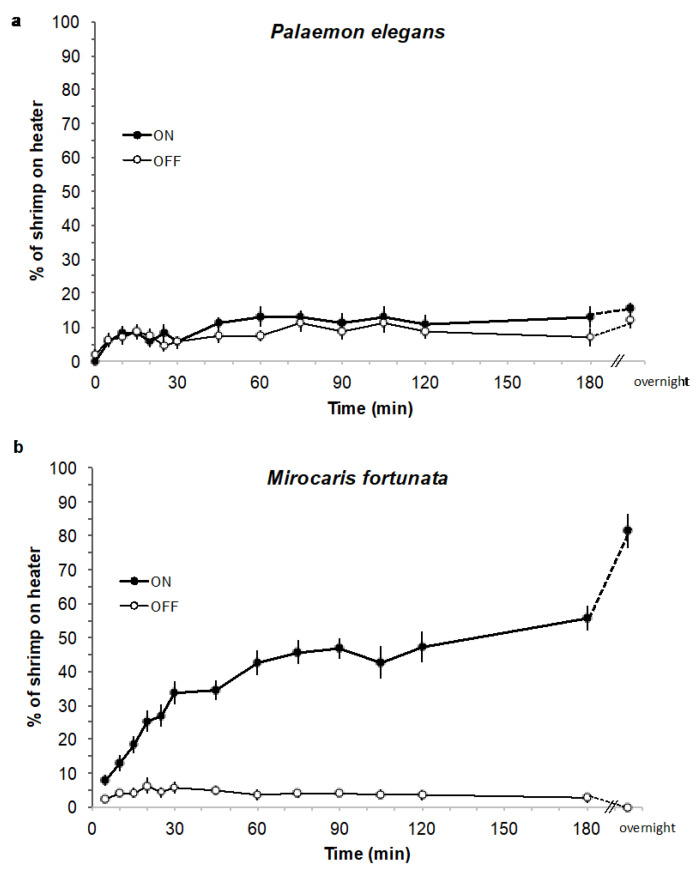
Two-choice experiment (on *vs* off heating thermostats) on *Mirocaris fortunata* and *Palaemon elegans*. Distribution of *M. fortunata* (**a**) and *P. elegans* (**b**) on the on and off thermostats over time. Two batches of 28 and 19 *M. fortunata* were tested 6 times each (*n* = 12 replicas per point, except overnight, *n* = 4 replicas). Two batches of 20 *P. elegans* were tested 4 times each (*n* = 8 replicas per point, except overnight, *n* = 4 replicas). The thermostats either on (set to 25 °C) or off were introduced on each side of the rearing tank (9 °C) in the upper region. The shrimp were observed for 30 min. The on and off thermostats were inverted between two consecutive trials. The distribution is presented as mean % individuals (± S.E.M.) on each thermostat.

**Table 1 insects-12-01043-t001:** Responses to odor and temperature stimuli for hydrothermal and coastal shrimp species. +, attraction; -, no attraction observed; nt, not tested. The food odor source can be a mussel extract or a shrimp food extract. The concentration of sulfide (Na_2_S) is 2 mmol L^−1^. The temperature source is approximately 17 °C in ambient water at 9 °C.

Stimulus	Hydrothermal Species		Coastal Species
	*Mirocaris fortunata*	*Rimicaris exoculata*	*Palaemon elegans*
Food	-	nt	+
Food + Sulfide	-	nt	+
Sulfide	-	-	-
Temperature	+	+	-

## Data Availability

On request to the authors for correspondence.

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
