# Peer review of "Do Hydrothermal Shrimp Smell Vents?"

_insects, 2021, doi:10.3390/insects12111043_

Round 1

Reviewer 1 Report

Ravaux et al. report results of several experiments investigating behavioural responses of the hydrothermal vent shrimp, Rimicaris exoculata and Mirocaris fortunata, to environmental cues (sulphide, food odour, temperature) characteristic of vent fluid and the vent field environment to assess the use of olfaction and thermal sensitivity for orientation in this dark and challenging habitat. The shallow-water shrimp, Palaemon serratus, is used as a non-vent living comparison. The manuscript first reports observations of grooming behaviour and SEM images of bacterial fouling of antennae and antennules. Then several experiments investigating attraction to sulfide and food odours with different experimental setups. Finally, experiments assessing thermotaxis are reported. The paper comprises experiments conducted over a number of years both on board oceanographic research cruises and in laboratories on dry land. Experiments represent a considerable effort, no doubt under challenging conditions, and make use of state-of-the-art equipment for maintaining deep-sea fauna under native pressure, enabling behaviour assessment under deep-sea conditions.

I have only minor comments for the authors to address (see below), which primarily concern the clarity of writing and presentation of the authors’ work.

Minor comments:

  1. Introduction

Both the ‘Simple summary’ and abstract focus on shrimp olfaction, no mention of temperature is made until the final sentence. Please introduce the concept that temperature could be used for orientation in hydrothermal vent fauna and that this was tested.

Line 21: suggest changing to ‘Whilst hydrothermal vent shrimp have been shown to possess…’

Line 22 suggest changing to ‘…none of our test setups allowed us to demonstrate…’

Line 68: change ‘mecanoreception’ to ‘mechanoreception’, here and throughout MS

Line 80: reference previous work [22-25]

  1. Materials and Methods

Line 175: please state vent mussel species name

Line 197-201: this section could be clearer. Perhaps state n for individual experiments and total n. For example, the 4 grooming experiments consisted of 3 batches of 3 adult and 1 batch of 9 juvenile R. exoculata, therefore n=18 shrimp.

Figure 2: Please change the use of letters for the different elements in this figure. For example, “c” is used for the diagram (top row, third from the left) and the corresponding photograph (bottom panel, forth from the left); the latter should perhaps be “c1” or “c’”. It is also used to label the camera in diagrams b and c.

Line 280: states 1 batch of 8 juveniles whilst line 198 states n=9 juveniles

Lines 257-267: please state the number of M. fortunata used

Line 260: the sulphide pulse in Figure 7a appears to be 6min

Line 265: data points in Figure 7a are plotted every minute. Please make it clear here that counts were done every minute.

Line 277: I suggest rephrasing this sentence as the experiment was done during BICOSE 2014 (line 268), but sulphide concentrations selected appear to be based on measurements taken during BICOSE 2018, ~4years after the experiments. This is similar for M. fortunata experiments done in 2013 and referencing a paper published in 2015 [38] (lines 262-265).

Line 280: please make it clear that counts were done every minute

Line 300-310: please state the number of shrimp used. Also, which experiments used gels with or without pH adjustment?

Line 322: please state that counts were done every minute

Line 360: change ‘gaze’ to ‘gauze’, here and throughout

Line 371: ‘Both species were starved for at least 48 h.’ Prior to the start of the experiment?

Line 387: change ‘pelets’ to ‘pellets’ (also line 398)

Line 396: change ‘heating’ to ‘heater’

Lines 410-423: was the ambient temperature in the tank constant? i.e. 9oC for the duration of the experiment or did the temperature increase over the experiment because of the heater?

Line 411: change ‘and’ to ‘or’

  1. Results

Line 445: experimental condition (temperature and pressure) are not labelled in Figure 4. Doing so would improve clarity.

Figure 4: y-axis ‘,’ is used rather than ‘.’ for decimals, please change. Also, the details of individual number and batch number stated in the legend are unclear, please revise.

Lines 459-479: I think that the use of repeated Student’s T-test may be inappropriate. Perhaps Two-way ANOVA is more appropriate here, to test grooming frequency between groups and appendage.

Line 565: change ‘neither’ to ‘either’

Line 568: suggest changing ‘A first experiment…’ to ‘An initial experiment…’

Line 603: change ‘aspect’ to ‘appearance’

Line 668: ‘…allowed to demonstrate…’, insert ‘us’? Or rephrase; e.g. ‘…allowed the demonstration…’

Figure 10b: a second y-axis with temperature could be added to each graph.

  1. Discussion

Line 744-746: I don’t understand the intended meaning of this sentence, please rephrase.

Line 789: change ‘…have led to suspect…’ to ‘suggest’

Line 875-882: based on your experimental setups, I don’t think you can draw robust conclusions about differences in species’ thermotaxis behaviour. R. exoculata were subjected to warm water pulses of 25oC and 10oC (those that elicited a response) in a 4oC tank whilst M. fortunate and P. elegans were subjected to a choice between two heaters (one ON and one OFF) in a 9oC tank. R. exoculata showed a rapid aggregation around warm water source; the temperature difference between warm water and ambient tank water was 21oC and 6oC. M. fortunata showed a gradual movement towards a warm spot on the ON heater, which varied between 0oC and 8oC warmer than the ambient tank water. The speed with which M. fortunata aggregates around a warm water pulse was not assessed, nor the speed with which R. exoculata may aggregate on a heater. Please amend this section to add an acknowledgement that differences in the speed of response to temperature and thermotaxis behaviour may be associated with differences in experimental setups.

Reviewer 2 Report

The manuscript “Do hydrothermal shrimp smell vents?” by Juliette Ravaux and colleagues presents new insights on the chemosensory behavior of deep-sea shrimp compared to coastal representatives. Beside behavioral assays investigating the impact of environmental cues like sulfide and food cues, as well as temperature, the authors also present evidence for grooming behavior in these deep-sea shrimp. Thus, this study adds some pieces to the puzzle of deep-sea life and the special adaptations to harsh habitats. The findings will be of importance for a broad readership, related to fields like for example ethology, ecology, and physiology.

Strength of the contribution: The authors provide many data on different aspects on such rare animals! The authors find an adequate balance in presenting their data without over-interpretation of results, which is very important because of the low sample size and low significance after data analysis.

Weakness of the contribution: Because of the many assays performed and the varying conditions and parameters, as well as species investigated, it is very difficult to follow the descriptions of setups, results, and interpretations (see details below).

In summary, I have no crucial concerns regarding the integrity of this contribution and think that it is well suited for publication in this journal. Nevertheless, I have some major issues, especially concerning the presentation of experiments/data and the interpretation of fouling degrees in deep-sea shrimp, which should be addressed in a revised version.

Major issues:

A: The manuscript contains a lot of different behavioral assays and there is a strong variation between similar experiments. I had a very hard time to figure out what exactly has been done, especially when reading the results and trying to connect it to the material and methods section. On the one hand, this issue is related to the different times and cruises when animals were collected. On the other hand, I think the structure and presentation of data can be improved. Here, I will give more detailed comments, with some suggestions (but not an exhaustive list) for improvement:

  • Lines 247 – 252: The duration between batches and juveniles varies from 53 minutes to 17 minutes. How is this discrepancy integrated in the results, especially figure 4? I think it is hard to compare the means of different durations, at least this issue must be clearly discussed while interpreting the data.
  • Lines 282ff: At this point it gets not clear, why you have combined sulfide and food experiments. You explain it later, but this information (reason, hypotheses) must be given here, in order to allow the reader to follow your thoughts. In these lines, it would be helpful to get an idea why you decided to perform so many different assays (two-choice, multiple-choice, single individuals, group experiments etc etc). Maybe you could expand table 1 adding sample size, hypotheses, time of experiment after collection etc. A well-structured table might help to guide the reader through that bunch of tests…
  • Lines 312ff: You explain the temperature pulses in minutes, but would it not be appropriate to add information about the volume of water, which has been introduced to the tanks in these 10 min pulses?
  • Figure 4 is hard to follow, because several important facts are not integrated: First, temperature and time differences between the batches of R. exoculata should be added. First, I wondered why R. exoculata has been treated in three batches, and P. serratus has pooled. It is difficult to follow without checking every time back to the M&M-section. In general, it would be beneficial to add sample sizes in all Figures. Further, it would be helpful to add the statistical results you obtained for that data set in the figure 4. Could you please add asterisks for significant test and NS for non-significant results?
  • Trying to follow your processed data, I was wondering if you will provide a raw data set, or at least a statement, how readers could obtain them?
  • Figure 7: Again, the changes in setups is quite hard to get. Here I am not sure, if the Figure is correct. I thought that the time investigated before the sulfide pulse is 5 min? However, in b) it is 10 min… Furthermore, the sulfide pulse had a duration of 10 min, but what happened to figure 7a, here its just 8 min? Duration after pulse has been investigated for ten min, but in 7a, it stops after 2 min after the pulse…. Please check and adjust.
  • Lines 613-615: Here, you mention, that you documented also the duration of contacts. If I am not wrong, this information is missing in the material and method section. Have you analyzed this parameter also for other assays? If yes, please give the results, if not, why not?
  • Figure 9: You had four choices in your test. One stimulus choice and three controls. How have you processed the data of controls? Is it the sum, or the mean of the three choices? That makes a big difference! If you take the sum, I do not get, why you have not just applied a two choice test?
  • Figure 10: Why does the data collection end after 15 min for the last pulse? You stated an investigation of 5 min before the pulse, 10 min during the pulse and 15 minutes after the pulse…

B: The authors discuss the different fouling levels between hydrothermal and coastal shrimp. I am wondering, if the authors do not miss a very important issue here, which can affect the outcome of the chemosensory assays and their conclusion. As far as I understood, the animals have been collected and cultured in tanks/aquaria with flow-through devices. Thus, the animals have been kept in surface water, correct? I think, that the microbiome of the water is an important factor, which can strongly affect the fouling degree. In this context, I remember some observations taken from Remipedia, which were collected in their natural habitat and transferred to aquaria. After a few days/weeks, animals started to produce mucus and showed altered grooming behavior (Koenemann et al., 2007, Journal of Crustacean Biology). If animals were kept in cave water, they survived longer without such dramatic behavioral changes.

Maybe a similar affect can be observed here? This would fit into the observation that coastal specimens do not show such high fouling degrees, as they are kept in “familiar” waters? I have doubts, that chemosensory sensilla can be kept functional with such a thick layer of bacteria on their pores. Maybe that’s the reason why you have not measured a reaction to the stimuli (at least food)?

You mention that grooming is more pronounced directly after collection and that this behavior gets less as longer they were held in the tanks (figure 4). Maybe this is a physiological reaction to pressure changes? Animals are not as fit as under natural pressure conditions due to decompression issues? And less grooming leads to higher fouling degrees?

In this context, it might be important to check when the antennae have been investigated in previous studies (lines 731-732, citations 22, 23). If specimens where kept certain time after collection of tissue, it might be not surprising that antennae are covered with bacteria. Related to this: Have the antennae used for electrophysiological experiments been checked for fouling as well (lines 756-759, citation 24)?

C: One issue that only receives attention in the final sentence of the manuscript is the possible role of sociality. While reading through the setup descriptions as well as the results, I was wondering if the number of individuals tested in the same time frame might affect the results, e.g., animals are observing and copying behavior of conspecifics. This issue needs a bit more attention in the interpretation of data.

Minor issues:

Line 32: Please also mention the names of the coastal species. In general, it gets not really clear, why the authors have chosen to investigate Palaemon elegans and P. serratus. Is there a specific reason, like e.g. a close phylogenetic relation or is it just because of availability reasons? Be it as it may, this should be specified somewhere in the main body of the manuscript.

Lines 47-52: The introduction of chemoreception is kind of unprecise. Chemoreceptive sensilla are not restricted to the first and second pair of antennae, but are distributed all over the body. In this context, it is important to remember that chemosensation does not only cover olfaction, but also gustation. Please try to be more precise and rephrase.

Line 68: In my opinion, the correct English term is mechanoreception, not mecanoreception. Please check throughout the manuscript.

Line 74: I always struggle about phrases like “short distance” and “long distance”, because it is too vague in my opinion. Could you please give examples what defines short and long distance in this context (see also Figure 1b, Detection to stimuli, or line 112)?

Line: 79: “alvinocarididae” and “palaemonidae” should be changed to “alvinocaridid” and “palaemonid”

Figure 1:

Panel 3: I have the feeling that this scheme is kind of redundant in its current form. For someone who is not familiar with the brain of arthropods/shrimp, it shows only some circles in a specific arrangement. For someone who is familiar with these brains, it does not contain any relevant information. Maybe, the authors can show a more complex scheme, including outline of the brain, for example as presented in Figure 3 from Machon et al., 2019, eLife?

Key results: Could you please add citations in the text? Although they are given in the figure legend, it would be nice to have them related to the specific statements.

3 Anatomy of the brain: “Similar lobe volumes”, I guess this statement is based on relative size? “Higher integrative centers more developed”: First, there are various higher integrative centers, e.g., hemiellipsoid bodies or the central body, please specify. And be more precise, what means “more developed”? Are you referring to volume as well?

Line 132: What is an “active behavior”?

Line 164: Please add coma after “Alternatively”.

Line 166: Please change “behavior” to “behavioral”.

Introduction/Figure 1: One suggestion, which might be beneficial for the story of the manuscript: It might be worthy to outline briefly adaptations of the olfactory neuropils associated to the antennules in other deep-sea or blind (cave) crustaceans/arthropods. For example, Remipedia (Stemme et al., 2012, 2016) or Cephalocarida (Stegner and Richter, 2011) show extraordinary large olfactory neuropils. Other blind crustaceans have comparably small olfactory centers (discussed in e.g., Stegner et al., 2015). As the olfactory lobes of the vent shrimp are not larger than those of coastal representatives, they might not rely that strong on chemosensory cues (although this might be expected due to the low light levels in the deep-sea habitat), which fits well in the results of the manuscript.

Figure 2: For better discrimination, I would term the “c” and “d” images in the lower line “c1” and “d1”.

Figure 6: The black lettering is hard to read. Maybe white letters with a black outline might help.

Figures: Please change format of scientific species names in the figures to italic style.

Lines 582 and 586: “and” is doubled in both lines.

Line 659: “The individual indeed…” should read “The individuals indeed…”?

Round 2

Reviewer 2 Report

In the revised version of the manuscript “Do hydrothermal shrimp smell vents?” the authors made a great effort and addressed most issues in a satisfactory manner. Nevertheless, I spotted some issues, which are listed below. After consideration of these issues (especially the last point), I strongly recommend this contribution for publication.

Line 127: Please change “central system” to “central nervous system”

Line 128: Please change “crustacean” to “crustaceans”

Lines 127-135: The statement of the authors might be a bit misleading. In its current form, it implies that species living in the dark have generally large well-developed olfactory neuropils. However, Stegner and colleagues found also small olfactory centers in blind cave-peracarids. Thus, that would fit well in the results by the authors, and the vent shrimps have the special attribute of large second order integration centers, which has not been documented for blind cave-forms. Further, Stegner etal 2015 have not worked on Remipedia, thus the authors might consider acknowledging the primary research on that topic, if referring to this crustacean group (General brain issues: Fanenbruck and Harzsch, 2004; Stemme et al., 2012; Higher integration centers: Stemme et al., 2016).

Line 661: Please change “appearence” to “appearance”.

Line 686: Full stop is doubled.

Line 833: Comma is doubled.

Line 1079: Please change “creustaceans” to “crustaceans”.

Figure 7: My initial point here has not been addressed (at least, I could not find any response or modification): In the M&M-section the authors mention that contact to pulse entrance was counted every minute for 5 min prior and 10 min after the 10 min-pulse (for M. fortunata in line 305, for R. exoculata in line 319). However, Figure 7 shows counting for only 1 min (?) after pulse in M. fortunata and 10 min prior to the pulse in R. exoculata… Please adjust M&M and/or the Figure.

Figure 9: Based on my initial comment on Figure 9, the authors responded that they have used the sum of control sites. I totally agree that four possibilities is more discriminant than just two. However, if the authors pool the data for the control, they lose this effect! If the authors would depict the percentage for all four choices perform an ANOVA analysis, there might be an effect even for the mussel extract in Fig. 9a and sulphide in 9d, as every choice needs to be treated separately.
